# When AI Benchmarks Plateau: A Systematic Study of Benchmark Saturation

**Mubashara Akhtar** [* 1 2]  **Anka Reuel** [* 3]

**Prajna Soni** [◇]  **Sanchit Ahuja** [◇ 4]  **Pawan Sasanka Ammanamanchi** [◇ 5]  **Ruchit Rawal** [◇ 6]  **Vilém Zouhar** [◇ 1]
**Srishti Yadav** [◇ 7]  **Chenxi Whitehouse** [◇ 8]  **Dayeon Ki** [◇ 6]

**Jennifer Mickel** [9]  **Leshem Choshen** [10]  **Marek Šuppa** [11]  **Jan Batzner** [12]  **Jenny Chim** [13]  **Jeba Sania** [14]
**Yanan Long** [15]  **Hossein A. Rahmani** [16]  **Christina Knight** [17]  **Yiyang Nan** [18]  **Jyoutir Raj** [5]  **Yu Fan** [1 19]
**Shubham Singh** [20]  **Subramanyam Sahoo** [21]  **Eliya Habba** [22]  **Usman Gohar** [23]  **Siddhesh Pawar** [7]  **Robert Scholz** [24]
**Arjun Subramonian** [5]  **Jingwei Ni** [1]

**Mykel J. Kochenderfer** [† 3]  **Sanmi Koyejo** [† 3]  **Mrinmaya Sachan** [† 1 2]  **Stella Biderman** [† 9]  **Zeerak Talat** [† 25]
**Avijit Ghosh** [† 26]  **Irene Solaiman** [† 26]

* Lead authors ◇ Top contributors † Advisors

This project was completed as part of the Evaluating Evaluations (EvalEval) Coalition: 🚩 https://evalevalai.com/

## Abstract

Artificial intelligence benchmarks are an important mechanism for measuring model progress and guiding deployment decisions. However, benchmarks quickly "saturate", making it difficult to differentiate models and diminishing their long-term value. In this study, we define *benchmark saturation* and analyze it across 60 language model benchmarks using 14 properties that relate to saturation. We find that nearly half of the our benchmarks exhibit saturation, with rates increasing with age. Further, we find that resilience to saturation is impacted by expert-curation, not by public test data. Our results suggest that design choices can extend benchmark longevity and inform more durable evaluation approaches.[1]

[1]ETH Zurich [2]ETH AI Center [3]Stanford University [4]Northeastern University [5]Independent Researcher [6]University of Maryland [7]University of Copenhagen [8]University of Cambridge [9]Eleuther AI [10]IBM Research, MIT-IBM Watson AI lab, MIT [11]Comenius University in Bratislava / Cisco [12]Weizenbaum Institute, Munich Center for Machine Learning, TUM [13]Queen Mary University of London [14]Harvard University [15]StickFlux Labs [16]AI Center, University College London [17]Scale AI Security and Policy Research Lab [18]Cohere [19]University of Hong Kong [20]University of Illinois Chicago [21]Berkeley AI Safety Initiative [22]The Hebrew University of Jerusalem [23]Iowa State University [24]Max Planck School of Cognition [25]University of Edinburgh [26]Hugging Face. Correspondence to: Mubashara Akhtar <mubashara.akhtar@ai.ethz.ch>, Anka Reuel <anka@cs.stanford.edu>.

*Proceedings of the 43rd International Conference on Machine Learning*, Seoul, South Korea. PMLR 306, 2026. Copyright 2026 by the author(s).

[1]Data and code are available in the Github repository.

## 1. Introduction

Artificial Intelligence (AI) benchmarks play a central role in measuring model progress, guiding deployment decisions, and informing policy and regulation (Hardy et al., 2025; 2024; Alzahrani et al., 2024; Union, 2024). Their value depends on their ability to distinguish between models. Yet many widely used benchmarks (*e.g.,* HumanEval (Chen, 2021)) have rapidly "saturated" (Maslej et al., 2024), with top-performing systems achieving near-identical scores. When performance converges within a narrow range, benchmarks lose discriminative power and provide limited guidance for model comparison or selection (Ott et al., 2022; Chen et al., 2025). Similar dynamics have been observed in other domains–for example, ImageNet (Deng et al., 2009) exhibits near-ceiling performance for most new models.[2]

Despite its importance, *benchmark saturation* has received limited systematic study. Prior work often notes performance plateaus, increased robustness (Ashury-Tahan et al., 2026a) or introduces new benchmarks in response (Wang et al., 2024b; Jimenez et al., 2024), but rarely analyzes the mechanisms driving saturation. It remains unclear why some benchmarks saturate quickly while others retain discriminative power, and there is no agreed-upon operational definition–whether saturation reflects near-human performance, fixed ceilings, or the loss of statistical separability among state-of-the-art models. We address these gaps by defining saturation as the loss of reliable discriminative power among top-performing models and operationalizing

[2]We want to emphasize that saturation of benchmarks is not always negative–if the benchmark was valid (Salaudeen et al., 2025), saturation means that a task can be considered "solved".

it through an uncertainty-aware saturation index derived from leaderboard data. Using this framework, we analyze 60 widely used text-based LLM benchmarks across domains and evaluation settings, annotated along dimensions such as task design, linguistic scope, data construction, and accessibility to study factors associated with saturation.

This paper makes the following contributions:

- We define benchmark saturation as the loss of reliable discriminative power among state-of-the-art models and introduce a reproducible, uncertainty-aware saturation index derived from leaderboard data.

- We identify which benchmark properties are systematically associated with saturation based on an analysis of 60 benchmarks. We find that commonly assumed safeguards, such as private test sets or closed-ended formats, have limited impact on saturation, while benchmark age and scale strongly predict it.

- We derive practical recommendations for benchmark design and lifecycle management, including monitoring practices, uncertainty reporting, and criteria for benchmark retirement or revision.

The remaining paper is organized as follows: Section 2 formalizes benchmark saturation and introduces our saturation index; Section 3 outlines benchmark collection and annotation; Section 4 presents the empirical analyses; Section 5 discusses implications and actionable recommendations; Section 6 concludes with limitations and future directions.

**Conflict of Interest Disclosure** This work was conducted as part of a research coalition, some of whose members (including coauthors) have contributed to models or reported evaluations analyzed in this study. All artifacts were subject to the same inclusion and annotation procedure, regardless of author involvement.

## 2. Conceptualizing Benchmark Saturation

In this section, we formally define benchmark saturation, introduce our uncertainty-aware saturation index, and analyze its robustness to key parameter choices.

### 2.1. Definition and Scope

We define *benchmark saturation* as the loss of reliable discriminative power among top-performing models under comparison. A benchmark is saturated when top-performing models cannot be statistically distinguished and performance approaches the empirically observed ceiling of the benchmark. This notion corresponds to what prior work informally describes as *performance saturation*–a plateau where inter-model differences become negligible (Justen, 2025; Wang et al., 2024b; Ott et al., 2022).

**Human performance ceiling.** Unlike definitions based on reaching human-level performance (Gupta et al., 2025), our definition does not rely on human baselines, which are often impossible to comprehensively obtain, unavailable, or inconsistently measured (Wei et al., 2025). Moreover, human-level performance does not imply saturation, as models may still be statistically distinguishable even after reaching human-level scores, allowing the benchmark to retain discriminative power. Previous analyses describe saturation patterns descriptively (Ott et al., 2022) or emphasize lifecycle management (Hardy et al., 2024), but do not provide a quantitative criterion to determine saturation.

**Saturation vs. stagnation.** We therefore formalize saturation as a measurable property derived from leaderboard uncertainty. We further distinguish *stagnation* from saturation: stagnation refers to statistical indistinguishability among top models, whereas saturation additionally requires that performance is near the empirical ceiling. In practice, limited noise estimates blur the distinction between the two.

---

**Definition: Benchmark Saturation**

A benchmark is *saturated* if the evaluated models can not be reliably distinguished by their performance scores and any further improvements are not statistically distinguishable under the evaluation protocol.
Formally, saturation is characterized by:
(1) statistically alike performance among different top-performing models
(2) top performing models are approaching the benchmark's empirically inferred ceiling.

---

If only condition (1) holds, we refer to the benchmark as being *stagnated* rather than saturated. In this case, observed indistinguishability may arise from model-level limitations, evaluation noise, insufficient benchmark sensitivity, or artifacts in the benchmark itself (e.g., spurious correlations or repetitive patterns) and may be overcome by future architectural, training, or evaluation advances. It is often difficult to clearly distinguish stagnation from saturation, as reliable estimates of evaluation noise and benchmark ceilings are rarely available.

Our operationalization should satisfy four desiderata:

1. **Model-relative:** Defined with respect to top-performing models at a given time.

2. **Metric-agnostic:** Applicable across common metrics (accuracy, F1, BLEU).

3. **Data-driven:** Avoids reliance on externally curated performance ceilings.

4. **Reproducible:** Produces identical decisions given the same leaderboard snapshot.

To formalize this notion, we consider the performance of a set of top-performing models on each benchmark. For

a given benchmark, let $s_1 \geq \cdots \geq s_k$ denote the scores of the top $k$ models (default $k = 5$). We introduce $k$ as a general parameter to avoid fixing the number of models considered, and to flexibly define the set of top-performing models used to assess saturation. In our analysis, we fix $k = 5$ to ensure comparability across benchmarks. This choice reflects a practical trade-off: smaller values of $k$ can lead to unstable estimates, while larger values risk mixing frontier models with older or less relevant ones, particularly given incomplete leaderboard coverage. Empirically, most benchmarks in our dataset report on approximately 5–7 recent, highly capable models, making $k = 5$ a reasonable and consistent choice.

### 2.2. Uncertainty-Aware Saturation Measurement

**Performance-based evaluation.** For accuracy-like metrics that are averages over a fixed test set of size $n$, we approximate the standard error of a model score $s$ as

$$\mathrm{SE}(s) \approx \sqrt{\frac{s(1-s)}{n_{\mathrm{eff}}}}. \tag{1}$$

where $n_{\mathrm{eff}} = n^{\alpha}, \qquad \alpha \in [0, 1], \text{ default } \alpha = 0.5,$

Note that accuracy-like metrics, metrics computed as averages over a fixed set of test samples with bounded per-sample contributions (e.g., accuracy, F1, BLEU) are broadly used. In such metrics, uncertainty can be approximated from finite-sample variability. For other metric types (e.g., Pass@k), the same framework is applicable but requires a benchmark-specific uncertainty estimate, such as bootstrap intervals or repeated-evaluation variance.

The effective test set size $n_{\mathrm{eff}} = n^{\alpha}$ down weights the nominal test set size $n$ to avoid an overly strong dependence of the saturation calculation on test set size. In our dataset, benchmark sizes vary substantially, ranging from a few dozen to several hundred thousand test samples, with a highly skewed distribution due to a small number of very large benchmarks. Using the raw test set size n would therefore cause the uncertainty term to be dominated by these outliers, leading to disproportionately small standard errors and artificially low saturation estimates for large benchmarks.

Thus, the standard error of the difference between the top model and $k$-th model is then

$$\mathrm{SE}_\Delta \approx \sqrt{\frac{s_1(1-s_1)}{n_{\mathrm{eff}}} + \frac{s_k(1-s_k)}{n_{\mathrm{eff}}}}. \tag{2}$$

Let $\Delta = s_1 - s_k$. We consider the top models to be statistically similar in performance if $\Delta \leq z \cdot \mathrm{SE}_\Delta$, where $z$ is a standard normal quantile (e.g., $z = 1.96$ for a 95% confidence level). This criterion considers both dataset size and evaluation noise. Evaluation uncertainty refers to the expected variability in leaderboard scores introduced by finite test set size and metric estimation noise. We define this uncertainty through the standard error of model scores and their differences, and treat performance differences within this range as statistically indistinguishable.

**Score compression.** To quantify to which degree performance scores at the top of the leaderboard are collapsing, we compute the normalized score range

$$R_{\mathrm{norm}} = \frac{s_1 - s_k}{\mathrm{SE}_\Delta}. \tag{3}$$

$R_{\mathrm{norm}}$ can be interpreted as a signal-to-noise ratio, comparing observed top-model score spread to expected evaluation uncertainty. Lower $R_{\mathrm{norm}}$ indicates greater saturation, with top-model differences falling within expected evaluation uncertainty and showing limited discrimination.[3]

**Empirical approximation of the noise ceiling.** Rather than assuming a fixed or externally defined noise ceiling, we treat the highest observed model performance ($s_1$) as an empirical proxy for the ceiling. Saturation is therefore assessed relative to the distribution of observed model scores, rather than with respect to an absolute performance target such as perfect accuracy (i.e., accuracy of 100%).

Strong clustering of top models at a low performance level should not be interpreted as the task being solved. Instead, such clustering indicates *model-level saturation*: the benchmark may no longer effectively distinguish between contemporary state-of-the-art models. However, as observed in prior benchmarks, this form of saturation reflects stagnation and does not preclude the benchmark from regaining discriminative power following paradigm shifts (*e.g.,* introduction of reasoning-centric or tool-augmented models) (Cobbe et al., 2021; Lewkowycz et al., 2022).

**Saturation index.** To capture saturation as a graded phenomenon, we combine the above signals into a continuous saturation index $S_{\mathrm{index}} \in [0, 1]$, which increases as top models become statistically indistinguishable. Benchmarks with higher values of $S_{\mathrm{index}}$ show stronger saturation evidence. We define the saturation index as

$$S_{\mathrm{index}} = \exp(-R_{\mathrm{norm}}^2), \tag{4}$$

which assigns high values when the performance differences are small relative to the evaluation uncertainty. High values of $S_{\mathrm{index}}$ indicate benchmarks where top-performing models are tightly clustered within evaluation noise, reflecting reduced discriminative power.

---

[3]In rare cases with near-zero uncertainty (e.g., deterministic near-perfect scores), we add a small $\epsilon$-stabilization in the denominator to avoid numerical instability.

| Setting comparison | Spearman correlation | Same bin (%) |
|---|---|---|
| $(k=3)$ vs $(k=5)$ | 0.92 | 48.3 |
| $(\alpha=0.5)$ vs $(\alpha=0)$ | 0.88 | 23.3 |
| $(\alpha=0.5)$ vs $(\alpha=1)$ | 0.92 | 18.3 |

*Table 1.* Sensitivity analysis of saturation index with respect to $k$ and $\alpha$. We report Spearman rank correlation and the percentage of benchmarks assigned to the same saturation bin.

For interpretability, we bucket benchmarks into five bins: *very low* ($< 0.01$), *low* ($[0.01, 0.3)$), *moderate* ($[0.3, 0.7)$), *high* ($[0.7, 0.9)$), and *very high* saturation ($\geq 0.9$). Notably, high saturation may also occur at lower absolute performance levels, reflecting model-level saturation rather than task-level completion. These bins are interpretable, empirically motivated ranges over a continuous score, intended to summarize broad saturation regimes rather than define strict thresholds. They reflect the spread of $S_{index}$ observed across benchmarks while preserving the index's continuity.

## 2.3. Sensitivity to Parameter Selection

We conduct a sensitivity analysis with varying $k \in 3, 5$ and $\alpha \in 0, 0.5, 1$ values. Across these settings, the resulting saturation indices remain highly correlated, indicating that the relative ranking of benchmarks is preserved. Table 1 gives an overview of correlation and the fraction of benchmarks that remain in the same bins. While we observe variation in bin assignments, most changes occur between neighbouring bins rather than large shifts, which suggests that the underlying signal is stable even when scores vary. We further observe that a smaller k-value (e.g., $k = 3$) increases variance due to limited model coverage, while larger $k$ risks mixing frontier and non-frontier models given incomplete, static leaderboard data. Similarly, $\alpha = 1$ leads to strong dependence on test set size, whereas $\alpha = 0$ ignores evaluation uncertainty. The choice $\alpha = 0.5$ is a balanced trade-off, moderating dataset size effects while preserving uncertainty awareness. Overall, absolute saturation values may shift slightly, but benchmark ordering remains stable.

## 3. Methodology

To study benchmark saturation, we combine structured benchmark annotations with leaderboard-based analysis.

### 3.1. Benchmark Collection and Annotation

**(1) Initial benchmark selection.** We used a three-stage, criteria-driven process to construct a representative benchmark set, focusing on benchmarks that (i) are actively used in contemporary LLM evaluation, (ii) provide sufficient longitudinal data, and (iii) vary along dimensions relevant to our hypotheses.

We compiled candidate benchmarks from two sources: *1.*

*Evaluation reports from major model developers.* We extracted benchmarks appearing in evaluation sections of official reports (such as model cards or technical reports) released between Jan 2022 and Nov 2025 by major developers, including OpenAI, Anthropic, Google, Meta, and Alibaba, to reflect real-world evaluation practices and downstream adoption. In total, we reviewed 61 documents and identified 190 benchmarks used in at least one report. *2. Highly-cited benchmark papers.* We additionally collected widely cited benchmarks via the Semantic Scholar API using keyword-based search (details in Appendix C).

**(2) Criteria-based filtering.** We filtered benchmarks to ensure suitability for analysis using the following criteria: *1. Public documentation:* Benchmark documentation (*e.g.,* paper, technical report, or website) must be publicly available. *2. Sustained usage:* Benchmarks extracted from developer reports must appear in at least five distinct reports to ensure broader relevance. *3. Clear evaluation protocol:* Benchmarks with ambiguous scoring, inconsistent splits, or unclear evaluation procedures were excluded. *4. Text-only scope:* We restricted our analysis to text-based benchmarks, excluding multimodal datasets to isolate language-related saturation effects. *5. Available leaderboard data:* We included only benchmarks with sufficiently up-to-date leaderboard data and multiple evaluated models, otherwise they were excluded (*e.g.,* BIG-Bench (Srivastava et al., 2023)).

**(3) Hypothesis-driven refinement.** We initially develop a set of hypotheses for potential causes of benchmark saturation (see Section 3 and Section B). To ensure adequate sample sizes across hypotheses, we supplemented the filtered set with benchmarks filling gaps along key dimensions (*e.g.,* multilingual, templated, open-ended). We conducted targeted Google Scholar searches using terms such as `AI benchmark`, `leaderboard`, `evaluation`, and `dataset`, combined with hypothesis-specific keywords (*e.g.,* `multilingual`, `open-ended generation`).

After filtering and refinement, the final dataset consists of **60 benchmarks**. See Table 3 for the full list.

**(4) Annotation protocol.** To test the hypotheses in Section 3, we annotated benchmarks according to the schema in Table 4. Annotations capture: (i) *temporality* (*e.g.,* release date), (ii) *saturation metrics* (*e.g.,* top-5 model scores), (iii) *data quality indicators*, (iv) *task structure* (*e.g.,* input/output format), and (v) *dataset properties* (*e.g.,* curation strategy). Annotations were collected through a structured protocol involving 23 researchers with expertise in dataset curation and evaluation. Each benchmark was independently annotated and secondarily reviewed using a predefined schema, followed by a final cross-benchmark consistency audit to resolve remaining ambiguities.

| Hypothesis | Statement |
| --- | --- |
| H1 | Public benchmarks saturate faster than private benchmarks with held-out test sets. |
| H2 | English-only benchmarks saturate faster than multilingual or mixed-language benchmarks. |
| H3 | Human-authored benchmarks are more resistant to performance saturation than synthetic or hybrid ones. |
| H4 | Benchmarks that use a closed-ended response format (e.g., multiple-choice, true/false) tend to saturate faster than those requiring open-ended generation. |
| H5 | Benchmarks that are older and more widely adopted saturate faster than newer or less-used benchmarks. |
| H6 | Non-templated benchmarks are more resistant to performance saturation than templated benchmarks. |

*Table 2.* Hypotheses on factors driving benchmark saturation.

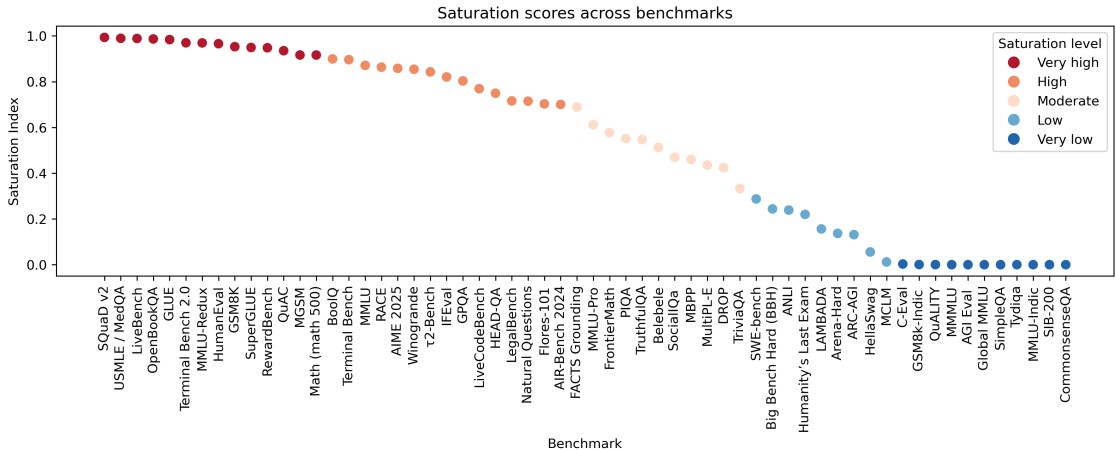

*Figure 1.* Overview of saturation scores across all studied benchmarks, ranked by their saturation levels.

**Final benchmark set.** Our benchmark selection spans a broad range of evaluation settings, including knowledge and reasoning tasks, multilingual, coding, long-context and factuality benchmarks, and recent agentic tasks. The benchmarks vary substantially in age (between 1 and 114 months), scale (from a few to hundreds of thousands of test samples), accessibility, output format, and construction style. Overall, the set includes 52 public and 8 private benchmarks, 44 English-only and 16 multilingual benchmarks, 28 closed-ended and 31 open-ended benchmarks, and 14 templated versus 46 non-templated benchmarks (Figure 1).

## 4. Empirical Analysis of Benchmark Saturation

We analyze saturation patterns across 60 text-based LLM benchmarks spanning domains, task formats, and evaluation settings. Using our saturation index (Section 2), we examine (i) saturation prevalence, (ii) temporal and exposure effects, and (iii) differences across benchmark properties.

### 4.1. Hypotheses-specific Analysis

We evaluate five hypotheses regarding potential drivers of saturation, grouped by accessibility ($H_1$), linguistic scope ($H_2$), data construction and quality ($H_3$), task design ($H_4$), popularity ($H_5$), and template ($H_6$); see App. B for details. Since benchmark age is itself positively associated with

saturation and differs across several benchmark categories, age is an important cofounding factor in cross-benchmark comparisons. We therefore distinguish age-balanced comparisons ($H_1$, $H_5$, $H_6$), where groups have similar maturity, from age-confounded comparisons ($H_2$–$H_4$) (Figure 2).

**Overall saturation patterns.** Saturation is widespread. Of the 60 benchmarks analyzed, 29 exhibit high or very high saturation ($S_{index} \geq 0.7$), out of which 14 fall into the very high category ($S_{index} \geq 0.9$). These benchmarks show strong compression among top-performing models, indicating limited discriminative power at the frontier. Across benchmarks, larger test sets are associated with lower saturation indices. Benchmarks with more test items show less score compression among top models, consistent with lower evaluation uncertainty and higher resolution. This relationship persists in joint regression (Section 4.2), suggesting that measurement scale impacts discriminative power.

**Temporal and exposure effects.** Figure 3 shows that the average saturation index increases with benchmark age. The proportion of saturated benchmarks rises from 42.9% for benchmarks released within the past 24 months to 54.5% for those older than 60 months, with corresponding mean $S_{index}$ values of 0.51, 0.52, and 0.60 across age bins. While the trend is modest and not statistically significant at conventional thresholds, it is directionally consistent: older benchmarks exhibit greater top-score compression. We eval-

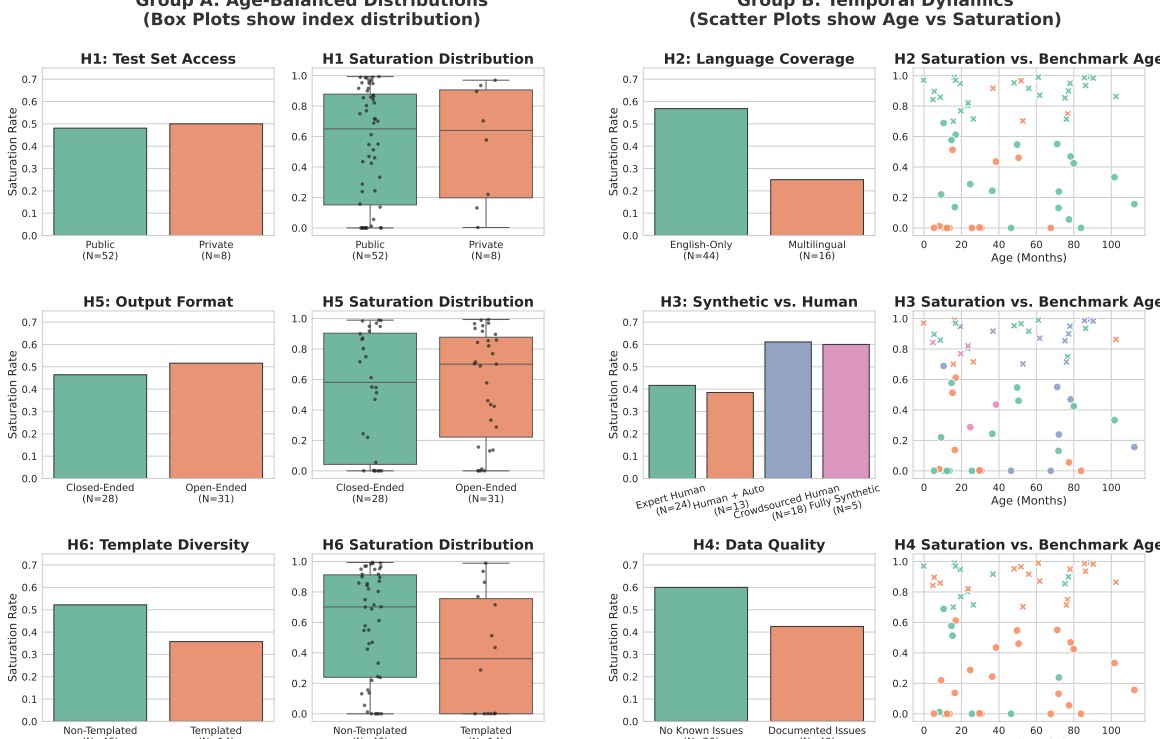

*Figure 2.* **Analysis of benchmark saturation** ($N = 60$). The figure is organized into two groups: **Group A (Left)** focuses on age-balanced categories ($H_1$, $H_5$, $H_6$), while **Group B (Right)** examines temporal dynamics ($H_2$, $H_3$, $H_4$), revealing that performance gaps in these categories are often driven by benchmark maturity. For each hypothesis, the first column of the group displays raw saturation rates. In the scatter plots, point colors correspond to the categories defined in the adjacent bar plots (legends omitted for brevity); $\times$ denotes saturated and $\circ$ denotes non-saturated benchmarks.

uate benchmark adoption using citation counts and inclusion in industry model release reports. Raw correlations show that benchmarks with higher citation counts tend to have higher mean saturation indices (Figure 4). However, after controlling for benchmark age, citation counts are not significantly associated with saturation ($\rho = 0.22$, $p = 0.12$). Citation growth rates ($\rho = 0.13$, $p = 0.37$) and frequency of appearance in technical reports ($\rho = 0.05$, $p = 0.73$) likewise show no significant association. These results suggest that maturity and cumulative exposure over time, rather than adoption metrics alone, better explain saturation patterns.

**Accessibility and task design.** Public ($N = 52$) and private ($N = 8$) benchmarks exhibit similar saturation distributions. We find no statistically meaningful difference in $S_{\text{index}}$ between the two groups. Hiding test data does not appear to prevent saturation once benchmarks are widely adopted, rejecting hypothesis $H_1$. Output format is age-balanced ($p = 0.40$). We observe no meaningful difference between closed-ended ($N = 28$) and open-ended ($N = 31$) benchmarks, suggesting that generation-based evaluation does not systematically preserve longer discriminative power.

**Benchmark composition and construction.** English-only benchmarks ($N = 44$) show higher raw saturation rates than multilingual ones ($N = 16$). However, benchmark age is a clear cofounding factor for $H_2$: multilingual benchmarks in our dataset are substantially younger on average (32.9 vs. 48.9 months). This indicates that the apparent robustness of multilingual benchmarks is largely explained by their young age rather than intrinsic resistance to saturation. Accordingly, we do not find support for $H_2$. We further examine whether benchmark design choices influence saturation, specifically whether expert- or human-curated benchmarks are more robust than crowdsourced or synthetic ones ($H_3$), and whether non-templated benchmarks are more resistant than templated benchmarks ($H_6$). Our analysis shows that curation categories differ significantly in age ($p = 0.0017$). Crowdsourced benchmarks are older on average and exhibit higher saturation rates in raw comparisons. Expert-curated benchmarks show lower saturation at comparable ages, and several of these benchmarks (*e.g.,* ARC-AGI, BIG-Bench Hard) remain unsaturated despite prolonged exposure. Furthermore, templated benchmarks ($N = 14$) do not differ significantly from non-templated ones ($N = 46$) in saturation behaviour ($p = 0.10$). Lit-

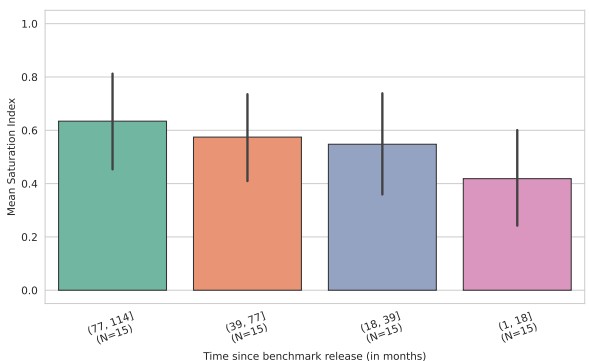

*Figure 3.* **Mean saturation index grouped by binned time since benchmark release (in months).** Older benchmarks exhibit higher average saturation, reflecting increasing performance compression among state-of-the-art models as benchmarks age. Error bars denote one standard deviation within each bin.

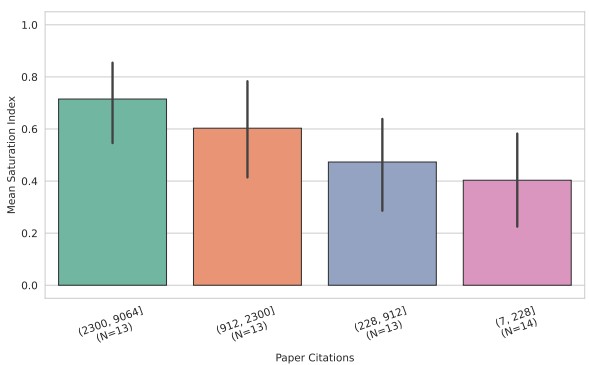

*Figure 4.* **Mean saturation index grouped by binned benchmark citation counts.** Benchmarks with higher citation counts exhibit higher saturation rates, suggesting that benchmark adoption and exposure are associated with reduced discriminative power over time. Error bars denote one standard deviation within each bin.

eral diversity alone does not appear to determine longevity. Fully synthetic benchmarks currently exhibit low saturation but are also relatively recent, limiting causal interpretation. These results suggest that expert-driven and adversarial design may improve robustness to saturation, though age remains a cofounding factor.

Benchmarks with documented quality issues ($N = 40$) exhibit higher saturation rates than those without ($N = 20$), but they are also significantly older on average (51.5 vs. 30.9 months; $p = 0.01$). In our annotations, documented quality issues include evidence of contamination, train-test overlap, noisy or low-quality examples, mislabeling or answer errors, documented demographic or linguistic imbalances, and other benchmark-specific problems such as unstable evaluation setups, ambiguity, or missing context. The association is consistent with multiple explanations: artifact exploitation, improved construction practices over time, or increased scrutiny of older benchmarks. Observationally, we find correlation but cannot isolate directionality.

## 4.2. Joint Analysis of Saturation Factors

To quantify which benchmark properties jointly explain variation in saturation, we fit a Bayesian regression model predicting $S_{index}$ from benchmark age, test set size, adoption proxies, accessibility, output format, templating, language coverage, curation strategy, and documented quality issues. The fitted model achieves $R^2_{Bayes} = 0.884 \pm 0.012$.

Across specifications, benchmark age and test set size show the most consistent effects. Adoption metrics contribute modestly but are not robust once age is included. In contrast, accessibility (public vs. private), output format, and templating do not exhibit reliable associations with saturation. Overall, the results indicate that saturation is more strongly associated with maturity and measurement scale than with commonly assumed design safeguards.[4]

# 5. Synthesis and Implications

Our empirical analysis reveals a consistent pattern: benchmark saturation is primarily driven by structural exposure dynamics and measurement resolution limits, rather than by isolated design choices. While contamination, overfitting, and ceiling effects have been discussed independently in prior work (McCoy et al., 2019; Murahari et al., 2024; Schaeffer, 2023), our results clarify which factors systematically correlate with saturation across benchmarks.

## 5.1. Saturation as a Structural Phenomenon

Our empirical results indicate that benchmark saturation is primarily a structural consequence of exposure dynamics and measurement resolution, rather than isolated design flaws. Two variables emerge as the most consistent predictors: benchmark age and test set scale.

**Age and exposure-driven compression.** Older benchmarks exhibit higher saturation indices, even after controlling for adoption metrics such as citation counts or inclusion in technical reports. Once age is accounted for, these popularity proxies no longer show associations with saturation, suggesting that cumulative exposure, not popularity alone, drives convergence. Repeated optimization against a stable evaluation target progressively compresses performance differences among frontier models. Our results are consistent with this interpretation, since older benchmarks exhibit higher saturation, although our analysis does not directly identify the causal mechanism. Similar plateau dynamics have been discussed qualitatively in prior work (Ott et al., 2022). This exposure effect is consistent with known risks of familiarity and memorization. Publicly accessible benchmarks increase the possibility that evaluation data, or close

---

[4]See Section F in the appendix for further details.

variants, appear in training corpora (see $H_1$ and Section 4.1), as well as findings in the literature (Zhou et al., 2023b; Balloccu et al., 2024). However, in our analysis, private test sets do not systematically reduce saturation once benchmark age is considered, suggesting that privacy alone is not sufficient. Even without explicit leakage or contamination, our finding that older benchmarks exhibit higher saturation supports the broader mechanism that repeated exposure to fixed evaluation formats encourages benchmark-specific optimization, narrowing observable performance gaps over time.

**Test set scale and measurement resolution limits.** In our empirical analysis (Section 4), larger evaluation sets are consistently associated with lower saturation indices. This suggests that discriminative power depends critically on statistical resolution. When evaluation uncertainty exceeds true performance gaps, top models become statistically indistinguishable even if substantive differences remain. Smaller test sets accelerate this effect, as variance dominates observed score differences. Moreover, reliance on coarse aggregate metrics (*e.g.,* single accuracy scores) can mask residual behavioral variation across subskills or input types (Murahari et al., 2024; Saxon et al., 2024). Taken together with our finding that older benchmarks tend to be more saturated, these results suggest that saturation often reflects loss of *relative separability* among top-performing models rather than complete task mastery (which would be desirable; importantly, benchmark saturation is a neutral, not a negative phenomenon. It only becomes an issue if saturation does not reflect task mastery). Benchmark maturity increases optimization pressure, while finite evaluation resolution constrains the ability to detect incremental gains. Saturation therefore emerges from the interaction between cumulative exposure and statistical measurement limits, even in the absence of explicit contamination or fundamental capability ceilings.

## 5.2. Safeguards That Do Not Prevent Saturation

Although benchmark age emerges as the strongest factor and most consistently correlates with saturation, we test the remaining hypotheses to evaluate whether commonly assumed safeguards retain explanatory power once we take age into account. Our results show that, these safeguards do not show robust associations with saturation in our data.

**Private test sets.** *Benchmark creators should not rely on private or held-out test sets alone as a long-term defense against saturation.* In our H1 analysis, we observe similar saturation distributions and no statistically meaningful difference in $S_{\text{index}}$ between public and private benchmarks. While contamination and memorization are well-documented risks (Zhou et al., 2023b; Balloccu et al., 2024; Deng et al., 2024; Sainz et al., 2024), secrecy alone does not prevent compression once distributional characteristics become widely known. Direct fine-tuning on evaluation data can trivially inflate scores (Schaeffer, 2023), but our results suggest that even without explicit leakage, prolonged exposure drives convergence.

**Open-ended output formats.** *Benchmark creators should not assume that switching from multiple-choice to open-ended generation alone will meaningfully extend benchmark usefulness over time.* In evaluating hypothesis H4, we observe no meaningful difference in saturation distributions between closed-ended (N=28) and open-ended (N=31) benchmarks. The output format comparison is age-balanced (p=0.40). Although multiple-choice benchmarks may enable overfitting strategies (Chandak et al., 2025), and models can exploit superficial cues (McCoy et al., 2019; Pacchiardi et al., 2024), format alone does not determine longevity. Compression seems to occur in both settings.

**Template diversity and multilinguality.** *Benchmark creators should prioritize refresh mechanisms, substantive difficulty and measurement resolution over surface-level diversity features such as templating or multilingual scope alone.* In evaluating hypothesis H6, we find that templated benchmarks (N=14) do not differ significantly from non-templated benchmarks (N=46) in saturation behaviour (p=0.10), suggesting that template diversity alone does not delay saturation. Multilingual benchmarks appear more robust in raw comparisons, but this effect is largely explained by recency. In evaluating hypothesis H2, we find that multilingual benchmarks (N=16) show lower raw saturation rates than English-only benchmarks (N=44), but this apparent advantage is confounded by benchmark maturity: multilingual benchmarks in our sample are substantially younger on average (32.9 vs. 48.9 months). While English-dominant pretraining corpora may accelerate ceiling effects on English-only tasks (Touvron et al., 2023; Wang et al., 2024a), age remains the dominant predictor.

Recent evidence from SWE-bench Verified also illustrates how plateaus can arise from evaluation artifacts rather than capability ceilings. OpenAI (2024) report that many frequently-failed tasks contain narrow or wide tests that reject functionally correct solutions and performance increasingly reflects training exposure to benchmark-associated repositories rather than general coding ability.

## 5.3. Structural Resistance to Saturation

A minority of benchmarks remain unsaturated despite substantial exposure. Qualitatively, these benchmarks tend to share structural properties that alter one or both of the above mechanisms. Benchmarks with adversarial or dynamic data collection (*e.g.,* Dynabench (Kiela et al., 2021)) reduce optimization stability by continuously updating the evaluation

distribution. Broad, capability-spanning initiatives such as BIG-Bench (Srivastava et al., 2023) expand coverage and limit narrow over-fitting. Holistic evaluation frameworks that track multiple behavioural dimensions (Liang et al., 2023) increase measurement granularity.

### 5.4. Implications for Benchmark Lifecycle Management

Our findings suggest that sustainable evaluation requires monitoring benchmark's discriminative power rather than relying on absolute score improvements. Benchmarks should be treated as evolving measurement instruments whose usefulness can reduce as models adapt to them.

**Benchmark design considerations.** Our findings suggest four actionable takeaways during benchmark design. *(1) Increase evaluation resolution.* Across our analyses, test set scale is one of the strongest predictors of lower saturation. Benchmark designers should therefore make sure that score differences between models exceed expected evaluation uncertainty. This can require larger test sets, harder examples, stratified reporting by subgroups of items (e.g., according to subskills), or multiple complementary metrics providing more insights into performance differences rather than a single aggregate score. *(2) Integrate dynamic benchmark updates.* Static benchmarks become easier optimization targets over time. Periodic refreshes, adversarial data collection, rotating hidden subsets, or continuously updated evaluation pools can reduce benchmark convergence resulting from exposure and prolong benchmark usefulness. *(3) Report uncertainty-aware statistics.* Integrate into leaderboard reporting confidence intervals, the spread of scores among top systems, and compression indicators in addition to aggregated peak scores. Small improvements that fall within evaluation noise should not be interpreted as meaningful progress. *(4) Define criteria for lifecycle management.* Benchmarks creators should include explicit revision, expansion, or retirement procedures during benchmark design once frontier systems become statistically indistinguishable, as highlighted in lifecycle-oriented evaluation frameworks (Hardy et al., 2024).

**When is saturation desirable?** Benchmark saturation is not inherently negative. If a benchmark is well-designed, valid, and measures a clearly defined capability, then convergence of top-performing models new the benchmark's ceiling may indicate genuine task mastery. In such cases, saturation reflects substantive progress: models can reliably perform the task the benchmark was intended to measure. However, saturation becomes problematic when it reflects loss of measurement resolution rather than capability completion. If score compression arises because evaluation noise exceeds true performance gaps, or because the benchmark lacks sufficient depth or coverage to differen-

tiate increasingly capable systems, then apparent convergence may mask unresolved weaknesses. In this scenario, models may appear indistinguishable despite meaningful differences in robustness, calibration, or generalization. The key distinction is whether saturation reflects true capability attainment or reduced discriminative power: the former signals progress, while the latter calls for revision or expansion.

## 6. Limitations and Future work

Our benchmark selection, though criteria-driven, reflects current evaluation practices and may overrepresent widely-adopted benchmarks. Top-N leaderboard snapshots may miss saturation dynamics for sparse or inconsistently evaluated benchmarks. The saturation index further depends on currently available frontier model evaluations, which may be incomplete, selectively reported, or inconsistently updated. We assume benchmark properties are time-invariant, yet attributes like annotation diversity evolve post-release. Similarly, benchmarks themselves may change over time through revised splits, refreshed test sets, or updated protocols, which are not captured in our static annotations.

Our analysis relies on publicly available leaderboard data, which posed several challenges. Multiple leaderboards may exist for a benchmark, often differing in evaluation setups (*e.g.*, LLM-judge prompts) and scoring criteria. Many leaderboards are not regularly updated and may omit newly released models. We therefore prioritized leaderboards based on visibility, recency, and result verification, though inconsistencies remain. Finally, our uncertainty estimates are designed for accuracy-like metrics over fixed test sets; metrics such as Elo ratings, pass@k, or judge-based evaluations require tailored variance estimates.

Future work should incorporate continuous-time leaderboard data and distinguish genuine saturation from temporary plateaus. Longitudinal analysis and causal studies comparing different exposure patterns could further clarify the mechanisms driving saturation. Studying performance shifts following major model innovations could clarify whether saturation is transient or persistent.

## 7. Conclusion

In this work, we present a systematic analysis of benchmark saturation. By introducing an uncertainty-aware saturation index and characterizing benchmarks across multiple design dimensions, we identify which properties are associated with saturation dynamics. Our findings challenge common assumptions (e.g. the protective role of private test sets) and highlight the importance of benchmark design, scale, and lifecycle management. This work provides a foundation for more robust and sustainable evaluation practices, designing benchmarks such that they remain informative over time.

## Impact Statement

Benchmark scores increasingly shape public discourse, model deployment, investment, marketing, policy decisions, and resource allocation in AI development. When saturated benchmarks are reported without appropriate context, they risk misinforming stakeholders about meaningful capability differences. Our analysis demonstrates that near-ceiling scores often fail to discriminate between models in ways that matter for downstream applications. This work encourages more careful communication of evaluation results, particularly when such results inform decisions in high-stakes domains such as healthcare, education, and public services.

## Acknowledgements

We thank Siva Kantha Rao Vanama, Abhijit Ubale, Vijaya Kumar Reddy Palreddy, Shivaprasad Chitta, Sasikanth Kotti, Wm. Matthew Kennedy, and Alexander Hoyle who supported this project through annotation efforts, discussions, and comments on the paper draft.

Mubashara Akhtar was supported by the ETH AI Center through an ETH AI Center postdoctoral fellowship. Hossein A. Rahmani's effort was supported by the Engineering and Physical Sciences Research Council (EP/S021566/1). Vilém Zouhar gratefully acknowledges the support of the Google PhD Fellowship. Marek Suppa was funded by the EU NextGenerationEU through the Recovery and Resilience Plan for Slovakia under the project No. 09I02-03-V01-00029. Jan Batzner was supported by the Federal Ministry of Research, Technology, and Space of Germany [Grant Number 16DII131]. Yanan Long thanks the TPU Research Cloud for computational support. Anka Reuel was supported by the Stanford Interdisciplinary Graduate Fellowship. Sanmi Koyejo is partially supported by NSF 2046795 and 2205329, IES R305C240046, ARPA-H, the MacArthur Foundation, Schmidt Sciences, Stanford HAI, RAISE Health, OpenAI, Microsoft, and Google.

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

# A. Related Work

**Benchmark Design & Evolution.** The development of AI benchmarks has evolved alongside advances in models, with an increasing focus on broad coverage and rigorous design (Hardy et al., 2024; Liu et al., 2024). Recent efforts emphasize diversity of tasks and continuous updates: for example, the BIG-Bench project crowdsourced hundreds of tasks to test language models' breadth (Srivastava et al., 2023), and dynamic benchmarks like Dynabench introduced adversarial, ongoingly collected test data so that evaluation remains challenging as models improve (Kiela et al., 2021). New benchmark paradigms also expand how performance is measured. The Holistic Evaluation of Language Models initiative, for instance, treats evaluation as a "living" benchmark that is continuously updated and tracks multiple metrics (accuracy, calibration, fairness, etc.) across many scenarios (Liang et al., 2023). Additionally, researchers have proposed principled frameworks for benchmark construction to ensure that datasets, tasks, and metrics truly capture the targeted capabilities (Salaudeen et al., 2025; Liu et al., 2024; Subramonian et al., 2023; Raji et al., 2021).

**Issues in AI Evaluations.** Despite continual benchmark innovation, significant challenges persist in how we evaluate AI systems. Various works have highlighted fundamental evaluation pitfalls in AI evaluation: Data contamination, i.e., when test content appears in training, can artificially inflate scores. For example, Schaeffer (2023) demonstrated that directly fine-tuning on a test set yields nearly perfect accuracy. Gamability of benchmarks is another concern: models often exploit spurious correlations or annotation artifacts to get high accuracy without genuine understanding. For instance, McCoy et al. (2019) have shown that models rely on superficial cues (*e.g.,* lexical overlap or keyword hints) instead of robust reasoning, achieving "right for the wrong reason" performance that fails on stress tests. Finally, reproducibility remains a challenge in AI evaluation: Seemingly superior results frequently vanish under minor experimental changes (*i.e.,* simply altering random seeds or dataset splits can yield statistically significant performance fluctuations and inconsistent evaluation protocols or opaque reporting have further complicated fair comparison of models (Xue et al., 2023; Habba et al., 2025; Ashury-Tahan et al., 2026b)). In parallel to this work Ashury-Tahan et al. (2026a), shows that this brittleness is highly reduced with saturation. Finally, Ott et al. (2022) show that benchmark saturation is a common occurrence, potentially making them misleading indicators of progress once models overfit to test quirks rather than achieve substantive gains. Yet, Ott et al. (2022) neither quantitatively define benchmark saturation nor do the authors analyze the causes of such saturation, two gaps we fill in this work. Relatedly, there is increasing awareness that aggregate metrics such as accuracy, F1, or single-scale scores often fail to capture nuanced model behavior (Murahari et al., 2024). This coarseness can create a misleading sense of benchmark saturation – models may reach near-ceiling aggregate scores while still exhibit substantial variation across subskills or input types. This apparent saturation, driven by the insensitivity of aggregate metrics, obscures remaining weaknesses and limits the diagnostic value of benchmarks. Consequently, several works advocate for holistic evaluation frameworks, including hybrid scoring schemes (Liang et al., 2023) or even a new discipline of model metrology to formalize rigorous, fine-grained measurement practices (Saxon et al., 2024).

# B. Hypotheses

We investigate five hypotheses about factors driving benchmark saturation, grounded in prior literature and design challenges. To support this analysis, we annotated 60 LLM benchmarks with related properties such as task format, data curation, and known quality issues. These annotations, detailed in Sec. 3, enable empirical testing of the hypotheses (Sec. 4).

**(H1) Data Access and Test Set Exposure:** *Public benchmarks saturate faster than private benchmarks with held-out test sets.* When test questions are public, models often memorize or leak this content from their training corpora, yielding artificially high scores that do not reflect true generalization: Zhou et al. (2023b) demonstrate that if an LLM's pre-training data contains examples from an evaluation benchmark, the model's score on that benchmark is significantly boosted. Likewise, Balloccu et al. (2024) conducted a large-scale analysis of GPT-3.5 and GPT-4 and found they were exposed to approximately 4.7 million benchmark samples during training, which may explain why these models quickly achieve near-perfect scores on popular public tests. Deng et al. (2024) devised a protocol to probe contamination on knowledge benchmarks and found that GPT-4 and Claude could fill in missing parts of real test questions with unnaturally high accuracy, implying the models had internalized those test items. These findings support H1: because public benchmarks are easily scraped or overfit, top model scores on them often reflect memorization.

**(H2) Language Coverage:** *English-only benchmarks saturate faster than multilingual or mixed-language benchmarks.* English dominates the pre-training corpora of most models (often >85–90% of tokens) (Touvron et al., 2023; OpenAI, 2020). Wang et al. (2024a) observe that an LLM's ranking across languages correlates strongly with the proportion of that language

in its training data, where models consistently excel at English and a few other high-resource languages, but struggle as one moves to less-seen languages. Consequently, an English-only task can hit a performance ceiling quicker because the model's exposure to English makes the task easier in distribution. In contrast, a multilingual benchmark introduces linguistic diversity that challenges the model's weaker languages and forces more robust generalization (Hu et al., 2020).

**(H3) Data Curation Strategy:** *Human-authored benchmarks are more resistant to performance saturation than synthetic or hybrid ones.* Human-curated evaluations typically span a richer diversity of problems and deeper conceptual challenges, often including intentionally difficult or adversarially crafted questions that thwart simple pattern-matching: Das et al. (2024) found that LLM outputs risk repetitive formats and missing corner-case reasoning. Diversity and deliberate complexity introduced by humans make it harder for models to "solve" benchmark tasks by exploiting superficial regularities (Gill et al., 2025). By contrast, LLM-generated (synthetic) benchmarks tend to exhibit hidden structural patterns or stylistic biases that models quickly learn to exploit, yielding artificially high scores without commensurate gains in real capability (Gill et al., 2025).

**(H4) Task Output Format:** *Benchmarks that use a closed-ended response format (e.g. multiple-choice, true/false) tend to saturate faster than those requiring open-ended generation.* Closed-ended tasks constrain the output space, making it easier for models to guess or recognize the correct answer without full understanding. The underlying mechanism is that closed formats convert complex tasks into simpler classification problems: the model's job is reduced to selecting one of $N$ options, a setup amenable to elimination strategies, frequency biases, or even memorized question-option pairs. Moreover, closed-ended benchmarks typically have an inherent guessing baseline (e.g. 25% for 4-choice questions), so even an uninformed model starts at a higher performance floor. Recent work demonstrated that some MCQ benchmarks enable overfitting of models such that they pick the right option without even reading the question (Chandak et al., 2025). By contrast, open-ended prompts (where the model must generate a free-form answer, explanation, or output) vastly expand the solution space and typically require a deeper grasp of the problem.

**(H5) Benchmark Maturity and Popularity:** *Benchmarks that are older and more widely adopted saturate faster than newer or less-used benchmarks.* As benchmarks mature and become widely adopted by the research community, they are repeatedly used for model development, hyperparameter tuning, prompt engineering, and evaluation, increasing optimization pressure against the benchmark itself. Prior work has noted that performance on popular benchmarks often improves rapidly shortly after release and then plateaus as models converge on similar solutions (Ott et al., 2022). Moreover, widely adopted benchmarks are more likely to be included—directly or indirectly—in training data or evaluation pipelines, further accelerating score convergence. As a result, benchmark age and popularity may jointly contribute to saturation by increasing exposure and targeted optimization, even when absolute task difficulty remains unchanged.

**(H6) Template vs Non-Template:** *Non-templated benchmarks are more resistant to performance saturation than templated benchmarks.* Templated benchmarks generate data samples using predefined patterns, structures, or parameterized templates, often resulting in repeated surface forms with limited variation. While such designs enable scalability and controlled coverage, they can introduce regularities that models quickly learn to exploit. In contrast, non-templated benchmarks consist of more diverse, free-form instances that are less constrained by fixed generation patterns. We therefore hypothesize that templated benchmarks, due to their structural regularities and reduced diversity, are more prone to faster saturation compared to non-templated, free-form benchmarks.

## C. Semantic Scholar Benchmark Collection

We retrieved all benchmarks appearing in the most-cited research papers between 2022 and November 2025 using Semantic Scholar API and the following queries (50 per keyword): `language model evaluation`, `LLM benchmark`, `foundation model benchmark`, `language model benchmark` and `language model evaluation benchmark`. The Semantic Scholar API retrieves the most relevant papers within a given time period. We first retrieve 200 relevant papers per keyword and select the top 50 cited papers. After merging all retrieved papers and deduplicating, we identified 186 papers using these keywords. We excluded non-text-based benchmarks. This keyword-based search yielded 2 additional benchmarks that were previously absent from our collection.

*Table 3.* Benchmarks included in our analysis (N=60)

| Benchmark | Reference | Benchmark | Reference |
|---|---|---|---|
| AGIEval | (Zhong et al., 2024) | MGSM | (Shi et al., 2023) |
| AIME 2025 | – | MMLU | (Hendrycks et al., 2020) |
| AIR-Bench | (Zeng et al., 2024) | MMLU-Indic | (AI, 2024b) |
| ANLI | (Nie et al., 2020) | MMLU-Pro | (Wang et al., 2024b) |
| ARC-AGI | (Chollet, 2019; Chollet et al., 2025) | MMLU-Redux | (Gema et al., 2025) |
| Arena-Hard | (Li∗et al., 2024) | MMMLU | (OpenAI, 2024) |
| Belebele | (Bandarkar et al., 2024) | MultiPL-E | (Cassano et al., 2022) |
| BIG-Bench Hard | (Suzgun et al., 2023) | Natural Questions | (Kwiatkowski et al., 2019) |
| BoolQ | (Clark et al., 2019) | OpenBookQA | (Mihaylov et al., 2018) |
| C-Eval | (Huang et al., 2023) | PIQA | (Bisk et al., 2020) |
| CommonsenseQA | (Talmor et al., 2019) | QuAC | (Choi et al., 2018) |
| DROP | (Dua et al., 2019) | QuALITY | (Pang et al., 2022) |
| FACTS Grounding | (Jacovi et al., 2024; 2025) | RACE | (Lai et al., 2017) |
| Flores-101 | (Goyal et al., 2022) | RewardBench | (Lambert et al., 2025) |
| Global-MMLU | (Singh et al., 2025) | SIB-200 | (Adelani et al., 2024) |
| GLUE | (Wang et al., 2018) | SimpleQA | (Haas et al., 2025; Wei et al., 2024) |
| GPQA | (Rein et al., 2023) | SIQA | (Sap et al., 2019) |
| MedQA | (Jin et al., 2021) | SQuAD v2 | (Rajpurkar et al., 2016; 2018) |
| GSM8K | (Cobbe et al., 2021) | SuperGLUE | (Wang et al., 2019) |
| GSM8K-Indic | (AI, 2024a) | SWE-bench | (Jimenez et al., 2024) |
| HEAD-QA | (Vilares & Gómez-Rodríguez, 2019) | $\tau$-Bench | (Barres et al., 2025) |
| HellaSwag | (Zellers et al., 2019) | TerminalBench | (Team, 2025) |
| HumanEval | (Chen et al., 2021) | TerminalBench 2.0 | (Merrill et al., 2026) |
| Humanity's Last Exam | (Phan et al., 2025) | TriviaQA | (Joshi et al., 2017) |
| IFEval | (Zhou et al., 2023a) | TruthfulQA | (Lin et al., 2022) |
| LAMBADA | (Paperno et al., 2016) | TyDiQA | (Clark et al., 2020) |
| LegalBench | (Guha et al., 2023) | MCLM | (Son et al., 2025) |
| LiveBench | (White et al., 2025) | Winogrande | (Sakaguchi et al., 2020) |
| LiveCodeBench | (Zheng et al., 2025; Jain et al., 2025) | WMT | (Kocmi et al., 2022; 2023; 2024) |
| MATH-500 | (Hendrycks et al., 2021) | FrontierMath | (Glazer et al., 2024) |

## D. Field Definitions for Annotation and Examples

This appendix provides detailed tables describing the annotation schema and benchmark metadata used in our analysis (Table 4), along with example rows illustrating the collected saturation metrics (Table 6) and dataset properties (Table 5).

*Table 4.* Benchmark Annotation Schema. Each benchmark in our dataset is annotated with the following fields to enable systematic analysis of saturation dynamics.

| Field | Description |
|---|---|
| *Identification & Temporal* | |
| Benchmark | Name of the benchmark being analyzed |
| Released On | Publication date of benchmark paper or public release on platforms like HuggingFace/GitHub |
| Citations | Citation count for the benchmark paper |
| *Saturation Measurement* | |
| Saturation Metadata | Top-5 model names and scores used to determine saturation status |
| Recent Models Evaluated | Whether frontier models released in 2025 (e.g., Gemini-2.5, Qwen-3) have been evaluated |
| SOTA in Paper | Best-performing model and score reported in the original benchmark paper |
| *Data Quality* | |
| Dataset Issues | Known post-release issues: contamination, biases, mislabeling/noise, or other data problems |
| Issue Sources | Follow-up papers or reports documenting identified dataset issues |
| *Task Structure* | |
| Input Format | Task input type: QA (MCQ), Instruction (open-ended), Coding (unit-test evaluated), or Agentic (multi-turn/tool-use) |
| Output Format | Expected response format: MCQ (select option) or Free-form (open generation) |
| Metric | Primary evaluation metric: Accuracy, BLEU, LLM-as-judge, or task-specific |
| *Dataset Properties* | |
| Curation Method | How data was created: expert human, crowdsourced, LLM-generated, or programmatically scraped |
| Curation Notes | Additional details on data collection methodology |
| Languages | Languages included in the benchmark |
| Sample Count | Number of evaluation examples in the benchmark |
| Availability | Whether benchmark and ground-truth labels are publicly accessible |
| Literal Diversity | Whether prompts use templated structures (e.g., "What is the capital of ___?") vs. natural variation |

*Table 5.* Benchmark Dataset Properties (Example Rows)

| Benchmark | Input | Output | Curation | Lang. | Samples | Citations | Avail. | Templated |
|---|---|---|---|---|---|---|---|---|
| Math-500 | Instruction | Free-form | Expert human | EN | 500 | 2398 | Public | No |
| GPQA Diamond | QA/MCQ | MCQ | Expert human | EN | 564 | 1180 | Public | No |

*Table 6.* Benchmark Saturation Analysis (Example Rows)

| Benchmark | Released | SOTA (Paper) | Recent Eval | Top-5 Models & Scores | Issues |
|---|---|---|---|---|---|
| Math-500 | Mar 2021 | 6.9 (GPT-2) | Yes | o3: 99.2; Grok-4: 99.0; DeepSeek R1: 98.3; GLM-4.5: 98.2; Claude Opus 4: 98.2 | Contam. |
| GPQA Diamond | Nov 2023 | 38.8 (GPT-4) | Yes | Grok4: 87.7; GPT-5: 85.4; Gemini-2.5: 84.4; Claude-4.5: 83.4; GLM 4.6: 82.9 | None |

## E. Benchmark-Level Saturation - Overview and Case Studies

To complement our analysis, we provide benchmark-level case studies in Table 7 illustrating how the saturation index behaves across different benchmarks. These examples highlight how score compression, evaluation uncertainty, and dataset properties jointly determine whether a benchmark is saturated, stagnated, or remains discriminative. These examples show a

range of saturation indices from fully saturated benchmarks, where evaluation noise obscures all meaningful differences, to unsaturated benchmarks that retain strong discriminative power.

Math-500 (very high saturation, $S_{\text{index}} = 0.92$). The Math-500 leaderboard shows that top-performing models are tightly clustered within a 1.0-point range (98.2–99.2), which lies within the estimated evaluation uncertainty ($SE_\Delta = 0.0338$). This results in a low normalized range ($R_{\text{norm}} = 0.30$), indicating that performance differences are not statistically meaningful and the benchmark has lost discriminative power.

LiveBench (very high saturation, $S_{\text{index}} = 0.99$). Although designed to mitigate contamination through regular updates, LiveBench shows high score compression (range = 1.09) relative to its uncertainty ($SE_\Delta = 0.1028$), resulting a very low $R_{\text{norm}} = 0.11$. Notably, this occurs at moderate performance levels ( 79%), suggesting model-level stagnation rather than task completion.

LiveCodeBench (high saturation, $S_{\text{index}} = 0.77$). LiveCodeBench shows stronger separation among top models (performance range = 3.9), which results in a higher normalized range ($R_{\text{norm}} = 0.51$). While still showing score compression, it demonstrates that also dynamically constructed benchmarks can saturate when evaluation resolution is limited.

TruthfulQA (moderate saturation, $S_{\text{index}} = 0.55$). The TruthfulQA leaderbaord shows a wider spread among top models (range = 6.7), which exceeds evaluation uncertainty. This leads to meaningful differentiation ($R_{\text{norm}} = 0.78$), but partial clustering indicates early signs of convergence, which is consistent with the benchmarks age and exposure.

Humanity's Last Exam (low saturation, $S_{\text{index}} = 0.22$). This benchmark shows substantial separation among top models (range = 11.4), which exceeds uncertainty ($R_{\text{norm}} = 1.23$). Combined with its large test set and recent release, it retains strong discriminative power and shows now clear sign of saturation.

*Table 7.* Representative benchmarks illustrating different saturation regimes.

| Benchmark | $n$ | Range | $SE_\Delta$ | $R_{\text{norm}}$ | $S_{\text{index}}$ | Level |
|---|---|---|---|---|---|---|
| Math-500 | 500 | 1.00 | 0.0338 | 0.2955 | 0.9164 | Very high |
| LiveBench | 1000 | 1.09 | 0.1028 | 0.1060 | 0.9888 | Very high |
| LiveCodeBench | 1000 | 3.90 | 0.0761 | 0.5124 | 0.7691 | High |
| TruthfulQA | 817 | 6.70 | 0.0863 | 0.7766 | 0.5471 | Moderate |
| Humanity's Last Exam | 2500 | 11.40 | 0.0926 | 1.2309 | 0.2198 | Low |

## F. Further Saturation Analysis

This appendix presents additional results from the joint regression analysis, including posterior coefficient estimates (Figure 5) and model performance, to provide a more detailed view of the factors associated with benchmark saturation (Figure 6).

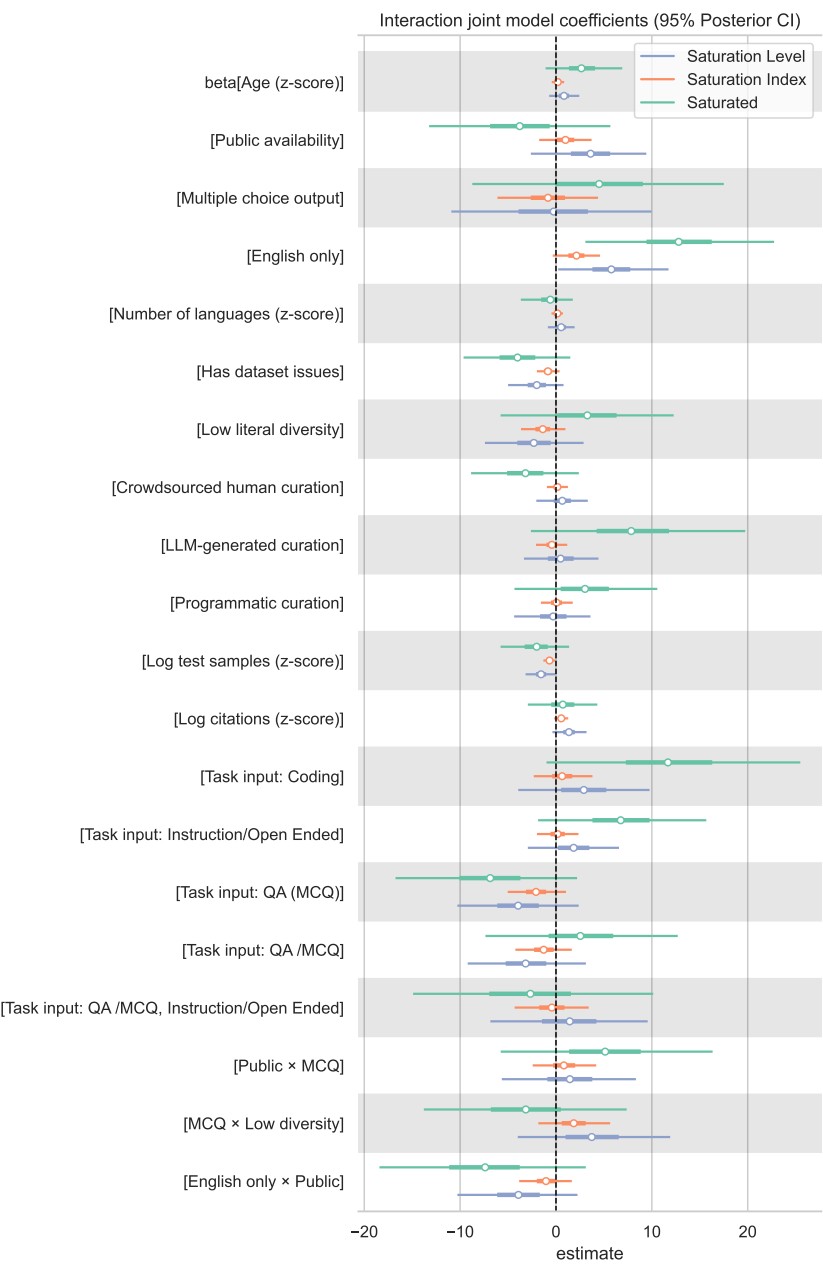

*Figure 5.* Forest plot of posterior regression coefficients from the joint interaction model predicting benchmark saturation. Points denote posterior means, inner line segments show 50% highest posterior density intervals, and outer segments indicate 95% credible intervals. Benchmark age and test set size exhibit the most consistent effects on saturation, while task format, literal diversity (templating), and their interactions show no strong effects after controlling for confounders.

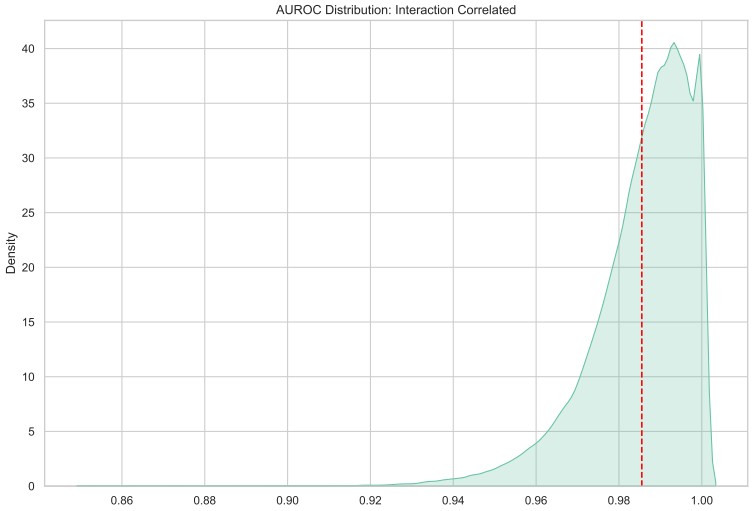

*Figure 6.* Posterior distribution of the AUROC for the interaction model predicting benchmark saturation. The distribution is tightly concentrated near high values (median approx. 0.98), indicating that the model distinguishes saturated from non-saturated benchmarks across posterior samples.

## G. Author Contribution Statement

| | |
|---|---|
| CONCEPTUALIZATION | M. Akhtar, A. Reuel |
| DATA CURATION | M. Akhtar, P. Soni, S. Ahuja, P. Ammanamanchi, R. Rawal, S. Yadav, C. Whitehouse, D. Ki, J. Mickel, M. Šuppa, J. Batzner, J. Chim, J. Sania, Y. Long, H. Rahmani, C. Knight, Y. Nan, J. Raj, Y. Fan, S. Singh, S. Sahoo, E. Habba, S. Pawar, R. Scholz, A. Subramanian, J. Ni, L. Struppek, A. Ghosh |
| INVESTIGATION | M. Akhtar, P. Soni, S. Ahuja, P. Ammanamanchi, R. Rawal, V. Zouhar, S. Yadav, C. Whitehouse, D. Ki, L. Ibrahim, J. Raj, Y. Fan, L. Struppek, U. Gohar, J. Mickel |
| METHODOLOGY | M. Akhtar, A. Reuel, P. Soni, S. Ahuja, P. Ammanamanchi, R. Rawal, V. Zouhar, S. Yadav, C. Whitehouse, D. Ki |
| SOFTWARE | M. Akhtar, P. Soni, S. Ahuja, P. Ammanamanchi, R. Rawal, V. Zouhar, S. Yadav, C. Whitehouse, D. Ki, J. Raj, M. Šuppa |
| FORMAL ANALYSIS | M. Akhtar, P. Soni, S. Ahuja, P. Ammanamanchi, R. Rawal, V. Zouhar, S. Yadav, C. Whitehouse, D. Ki, Y. Long, J. Chim, J. Sania, M. Šuppa, Y. Nan |
| WRITING (ORIGINAL DRAFT) | M. Akhtar, A. Reuel, P. Soni, S. Ahuja, P. Ammanamanchi, R. Rawal, V. Zouhar, S. Yadav, C. Whitehouse, D. Ki, R. Scholz, L. Ibrahim, Y. Fan |
| WRITING (REVIEW & EDITING) | M. Akhtar, A. Reuel, P. Soni, S. Ahuja, P. Ammanamanchi, R. Rawal, V. Zouhar, S. Yadav, C. Whitehouse, D. Ki, J. Mickel, L. Choshen, M. Šuppa, J. Batzner, J. Chim, J. Sania, Y. Long, H. Rahmani, C. Knight, Y. Nan, J. Raj, Y. Fan, S. Singh, S. Sahoo, E. Habba, U. Gohar, S. Pawar, R. Scholz, A. Subramanian, J. Ni, L. Struppek, L. Ibrahim, M. Kochenderfer, S. Koyejo, M. Sachan, S. Biderman, Z. Talat, A. Ghosh, I. Solaiman |
| VISUALIZATION | M. Akhtar, J. Raj, V. Zouhar, M. Šuppa, C. Whitehouse |
| SUPERVISION | M. Akhtar, A. Reuel, L. Choshen, M. Kochenderfer, S. Koyejo, M. Sachan, S. Biderman, Z. Talat, A. Ghosh, I. Solaiman |

