# OpenReview forum: "When AI Benchmarks Plateau: A Systematic Study of Benchmark Saturation"
_ICML.cc/2026/Conference — ICML 2026 regular_

### Official Review · Reviewer_9CPY · 2026-02-21

**Soundness:** 3
**Presentation:** 2
**Significance:** 3
**Originality:** 3
**Overall Recommendation:** 4
**Confidence:** 3

**Summary:**

This paper focuses on the benchmark saturation problem and analyzes 60 mainstream benchmarks from the perspectives of 14 properties, resulting in valuable insights for extending benchmark longevity and informing strategies for more durable evaluation. The authors formalize benchmark saturation as the loss of reliable discriminative power among state-of-the-art models and introduce a saturation index derived from leaderboard data.

**Compliance With Llm Reviewing Policy:**

Affirmed.

**Final Justification:**

The conclusion finally provided by the author is very compelling, namely, it indicates that LiveBench is already highly saturated. After carefully reading the other opinions and responses, in general I recognize the contribution of this article, but some aspects of the rationality analysis of the indicators and the presentation of the article still need improvement. Therefore, I raised the score to 4.

**Key Questions For Authors:**

1. Tab. 2 can be combined with Tab. 3 and placed in the appendix.
2. It is recommended to explain evaluation noise and insufficient benchmark sensitivity in line095 more clearly. Examples, quotations, or figures, are necessary, which can enhance the readability of the article.
3. Line138-141: What are the reasons for choosing these numerical ranges? Are they based on experiments or theories?
4. App. C and App. D lack textual descriptions.
5. The saturation analysis and numerical results for the benchmark individuals (e.g., livebench, livecodebench) are lacking. The current experiments mainly focus on the general analysis of the 60 benchmarks. It is suggested to include more charts, or to add specific benchmark labels to the scatter plot in figure 1.
6. Lack of the saturation degree score charts for 60 benchmarks.
7. I believe the thesis and research subject of this article are valuable. I will consider changing the score based on the author's rebuttal.

**Limitations:**

yes

**Strengths And Weaknesses:**

**Strengths:**

1. Extensive experiments and analyzes on 60 LLM benchmarks for the mechanisms driving saturation.
2. 14 properties spanning task design, data construction, and evaluation format, are considered.
3. The research question is of great significance, and the conclusion is well-supported by sufficient experimental evidence.

**Weaknesses:**

1. I think the layout and organization of the article can be further improved.
2. Lacking some theoretical support for the evaluating hypotheses.

---

> ### Author Rebuttal · Authors · 2026-03-31
>
> We thank the reviewer for the constructive review and are excited to read that they appreciate the significance of the problem, the scope of the collected data and study, and the strength of the experimental analysis.
>
> We appreciate the reviewer’s concrete suggestions regarding layout and structure of the paper. In the revision, we will make the following changes to improve clarity and readability of the paper:
>
> * We will add a brief “overview paragraph” at the end of the introduction to guide the reader through Sections 2–5.
> * To address the your comment on explaining evaluation noise and sensitivity we will include a new figure showing:
>   * score distributions
>   * overlap due to uncertainty
>   * resulting saturation signal
> * As pointed out, Fig 1 can be improved visually; therefore, we will:
>   * increase label and font size
>   * label some representative benchmarks in scatter plots (as you suggested)
>   * add a legend explaining symbols more prominently (in addition to the current explanation in the caption)
> * We will additionally include a visualization (e.g., a sorted bar chart or histogram) of saturation scores across all 60 benchmarks to provide a clearer overview of their distribution. Moreover, we have released the entire collected dataset (including saturation scores) and will add a link in the camera-ready version to enable readers to access and reuse the data.
> * We will merge Table 2 (the benchmark list) with Table 3 and move it to the appendix as suggested.
> * We will add clear textual descriptions for Appendix C and D, referencing the tables and figures that are part of the appendices.
> * We will also discuss a few representative benchmarks (e.g., LiveBench, LiveCodeBench) as case studies to provide more concrete interpretation of the results and help the reader understand and interpret the calculated saturation index, as well as other studies factors related to benchmark design.
>
> The bins are chosen as interpretable ranges over the continuous saturation index to provide a coarse but practical summary of saturation levels, rather than reflecting strict theoretical thresholds. They are empirically motivated (given the spread of s_index across all 60 analysed benchmarks) while considering the continuous nature of the saturation index. Thank you for raising this points, we will add details on the spread of scores and how the bins were selected in the revised paper version.
>
> In the review you pointed out that we lack “some theoretical support for the evaluating hypotheses.” - we are not sure we correctly understood this comment, but in case the reviewer thought this is missing - in Appendix A we have already listed and discussed the hypotheses and their motivation in detail. We can also move (a condensed version of) it to the main paper once we have more space for the camera ready version.

---

> > ### Author Rebuttal · Reviewer_9CPY · 2026-04-01
> >
> > The author merely gave an outlook-like response. Could some numerical results be provided for representative benchmarks?

---

> > > ### Author Response · Authors · 2026-04-02
> > >
> > > | Benchmark | n | n_eff | s1 | s5 | Range | SE_Δ | R_norm | S_index | Level |
> > >    |---|---|---|---|---|---|---|---|---|---|
> > >    | Math-500 | 500 | 22.36 | 99.20 | 98.20 | 1.00 | 0.0338 | 0.2955 | 0.9164 | Very high |
> > >    | LiveBench | 1,000 | 31.62 | 79.33 | 78.24 | 1.09 | 0.1028 | 0.1060 | 0.9888 | Very high |
> > >    | LiveCodeBench | 1,000 | 31.62 | 91.70 | 87.80 | 3.90 | 0.0761 | 0.5124 | 0.7691 | High |
> > >    | TruthfulQA | 817 | 28.58 | 91.10 | 84.40 | 6.70 | 0.0863 | 0.7766 | 0.5471 | Moderate |
> > >    | Humanity's Last Exam | 2,500 | 50.00 | 37.72 | 26.32 | 11.40 | 0.0926 | 1.2309 | 0.2198 | Low |
> > >
> > > To provide concrete examples of how the saturation index captures different regimes of discriminative power, we pick five representative benchmarks.
> > >
> > > - Math-500 (very high saturation, $S_{\text{index}} \approx 0.92$) is a benchmark that has lost discriminative power among frontier models: The scores of the top five models are statistically indistinguishable, as they span only 1.0 percentage point (98.2–99.2, with a standard deviation of 0.43). This narrow range falls well within the estimated evaluation uncertainty ($\text{SE}\Delta = 0.034$, $R\text{norm} = 0.30$). Math-500 illustrates the structural predictors identified in our analysis: it has a small test set ($n=500$), is 57 months old, and has known contamination issues.
> > >
> > > - LiveBench ($S_{\text{index}} \approx 0.99$) is a particularly notable example. Even though it was specifically designed to prevent contamination through monthly question updates, it shows the highest saturation in the chosen datasets after just 18 months. Absolute performance remains moderate (mean = 78.7%), suggesting model-level stagnation rather than task mastery. This is evidenced by the top-5 scores being tightly compressed into a 1.09-point range (78.2–79.3), with a small standard deviation of 0.41 and a low $R_{\text{norm}}$ of 0.11.
> > >
> > >
> > > - LiveCodeBench ($S_{\text{index}} \approx 0.77$) tells a related story: also built for contamination resistance through continuously sourced competitive programming problems, it reaches high saturation at 21 months with top-5 scores spanning 3.9 points (87.8–91.7, std = 1.38, $R_{\text{norm}} = 0.51$), though it retains slightly more discriminative power than LiveBench; both benchmarks use only 1,000 test items,  underscoring that contamination-resistant design is insufficient when test set scale limits statistical resolution.
> > >
> > >
> > > - TruthfulQA (moderate saturation, $S_{\text{index}} \approx 0.55$) shows a wider top-5 spread of 6.7 points  (84.4–91.1, std = 2.55, $R_{\text{norm}} = 0.78$) that exceeds the statistical indistinguishability threshold, so meaningful differentiation between systems remains possible, though early signs of convergence are visible at 51 months of age.
> > >
> > > - "Humanity's Last Exam" demonstrates strong resistance to saturation with a low saturation index ($S_{\text{index}} \approx 0.22$). Even at only 11 months old, its discriminative power remains high, with a top-5 score range of 11.4 points (26.3–37.7), a standard deviation of $4.45$, and a normalized range ($R_{\text{norm}}$) of $1.23$. This range significantly exceeds typical evaluation uncertainty. With 2,500 expert-curated questions and a modest mean top-5 score of $30.4%$, this test combines the qualities our analysis links to saturation resistance: recency, scale, and expert curation.
> > >
> > > These cases collectively demonstrate the range captured by $S_{\text{index}}$, showing a progression from benchmarks where evaluation noise entirely masks performance differences, through transitional phases, to those that remain distinctly informative.
> > > We will add these details to the camera-ready version.
> > >
> > > We want to thank the reviewer for the thoroughness of their review. Your comments will help us significantly improve our paper, thank you so much!

---

### Official Review · Reviewer_zhVz · 2026-03-12

**Soundness:** 3
**Presentation:** 3
**Significance:** 3
**Originality:** 4
**Overall Recommendation:** 5
**Confidence:** 4

**Summary:**

The authors perform a systematic study of AI benchmarks ability to differentiate models over time.  The saturation of benchmarks over time is studied as a phenomena, with five hypotheses evaluated on data from 60 benchmark datasets.  From the controllable characteristics of the benchmarks (14 total), and an output measure of saturation (carefully defined and distinguished from stagnation), the aggregate trends of benchmarks over time were analyzed empirically.  The authors shared their findings of what made for durable benchmarks, with the aim of helping the field to ensure future benchmarks can provide the most benefit to the development of AI capabilities.

**Compliance With Llm Reviewing Policy:**

Affirmed.

**Final Justification:**

The author's changes address my questions and clarifies the manuscript.  I will retain my accept recommendation, as I believe the community need for this original meta analysis of benchmarking outweighs the limitations of the paper noted in the reviews (methodological choices, limited depth of analysis across the multiple hypotheses, providing theoretical framework, etc.).  As the concept of model metrology matures, I expect that future work will have greater theoretical rigor; this paper is a welcome contribution on the current state of the field.

**Key Questions For Authors:**

Q1; The $s$ parameter in Equation 1 is only described as model score, from its use it in the equation, it is implied to be bounded between 0 and 1, is this the case?
Q2: Appendix C seems to be missing, is this in error?
Q3: The alpha parameter in Eq 2 seems to be introduced as a convenient way of dealing with the large test set sizes, but its effects on the study did not seem to be evaluated. Do the results of the study change with different default values?
Q4: Figure 1 and Section 4.1 seem to refer to a sixth hypothesis H6 not described in Table 1 or Appendix A, what is H6?

**Limitations:**

yes

**Strengths And Weaknesses:**

The authors are exploring an important topic to help ensure the responsible construction of future benchmarks for AI models, The work is significant in that it may push more researchers to seriously consider metrology as a way to make the most of benchmark creation efforts.  My main critique would be in the choice of analysis approach for the study, though I think the authors approach was adequate for the five hypotheses they explored.  The metrics of standard error seem to be derived based on standard error of the accuracy proportion, and then normalized to provide a convenient distribution of a saturation metric.  I would prefer to have framed the saturation of benchmarks as one of model discrimination (Box and Hill, 1967), treating each benchmark as an experiment with an expected gain of information that diminishes as the benchmark saturates.  Concepts like the empirical noise ceiling and model-level saturation could also be considered in a more principled way under this framing.  This would also allow for considering the performance of models on other benchmarks to determine when a specific benchmark saturates.   In terms of originality, I'm not aware of other work that explores the covariates of AI benchmark saturation in such detail, this was a highly informative study for me, and I see more opportunities in the future to better characterize the life cycle and utility of AI benchmarks.

---

> ### Author Rebuttal · Authors · 2026-03-31
>
> We thank the reviewer for the constructive review, and for recognizing the importance of the problem as well as the originality of our study and its potential impact on better characterizing the life cycle and utility of AI benchmarks. We appreciate the reviewer’s suggestions and agree that framing benchmarks in terms of information gains is a useful and complementary perspective. In our work, we focus more on the operationalization of saturation, i.e., when top-performing models become statistically indistinguishable and see this as a first step toward quantifying the saturation phenomenon.
>
> Below we respond to the raised questions:
>
> **Q1.** Yes, in our formulation s denotes a normalized score bounded between 0 and 1 (e.g., accuracy or similar metrics). Thank you for pointing out, we will clarify this in the paper.
>
> **Q2.** Thank you for pointing this out. Tables 4, 5 and 6 belong to Appendix C; we will add clear textual descriptions for Appendix C and D and reference the tables and figures that are part of these appendices.
>
> **Q3.** The parameter alpha is introduced to moderate the influence of test set size on uncertainty estimates. We agree that this should be better justified and will include additional explanation as well as a sensitivity analysis to confirm that our main conclusions are robust across reasonable values of alpha.
>
> **Q4.** H6 refers to the analysis of template diversity (templated vs. non-templated benchmarks). We will correct the inconsistency and ensure it is included in both Table 1 and Appendix A.
>
> Moreover, we agree that concepts such as empirical noise ceilings and model-level saturation are valuable extensions. However, estimating a reliable performance ceiling is challenging. Previous work often relies on human performance, which can be noisy and depend on factors such as annotator expertise and evaluation setup.
>
> Thank you also for the suggestion of considering model performance across multiple benchmarks to better understand when a specific benchmark saturates. Our framework could be extended in this direction by combining saturation estimates across benchmarks or tracking relative performance shifts over time. We will expand the discussion and highlight them as promising directions for future work.

---

> > ### Author Rebuttal · Reviewer_zhVz · 2026-03-31
> >
> > Thank you for the responses, my questions are resolved

---

### Official Review · Reviewer_7yBx · 2026-03-12

**Soundness:** 3
**Presentation:** 4
**Significance:** 2
**Originality:** 3
**Overall Recommendation:** 4
**Confidence:** 4

**Summary:**

The paper tackles an interesting question that's been floating around the community for a while: how do we know when a benchmark is actually "saturated"? The authors provide a formal, quantitative definition of saturation, distinguishing it from stagnation, and propose a continuous "Saturation Index" based on the statistical indistinguishability of top-performing models relative to evaluation noise.  They apply this to 60 text-based LLM benchmarks across 14 properties, test five hypotheses about what drives saturation (public vs. private, English-only vs. multilingual, human vs. synthetic, closed vs. open-ended, old vs. new). The key findings challenge conventional wisdom: benchmark age and exposure are the dominant predictors of saturation, while commonly assumed safeguards like private test sets, open-ended formats, and template diversity do not reliably prevent it. The paper concludes with practical recommendations for benchmark design and lifecycle management.

**Compliance With Llm Reviewing Policy:**

Affirmed.

**Final Justification:**

Thank you to the authors for the detailed rebuttal. I retain my assessment of 4 - Weak Accept.

**Key Questions For Authors:**

In summary, I really like the topic of the paper and the extensive experiments which demonstrate huge efforts devoted into the work. My main questions are around exploration of deeper causal mechanisms and actionable insights for benchmark creators. I would be happy to increase my score to 5 or 6 (if I'm convinced that the takeaways from the paper could be actionable guidelines rather than just interesting and counter-intuitive insights) if the following questions are adequately addressed.

**On causal mechanisms:**

(1) The paper establishes strong correlations between benchmark age/exposure and saturation, but the causal mechanism remains speculative. Is it memorization? Benchmark-specific overfitting? Distributional familiarity? Loss of novelty? Could you design an interventionist study (e.g., comparing benchmarks with similar age but different exposure patterns, or tracking how saturation changes after a paradigm shift like the introduction of reasoning models) to isolate the underlying cause? Without this, how can your findings guide benchmark designers beyond telling them to "make benchmarks newer"? This gets at the user's core concern about whether the paper addresses the "evaluation of evaluation" problem or merely describes it.

**On the effective sample size parameter:**

(2) Equation (2) introduces an interesting effective sample size formula and sets α=0.5 as a default for downweighting test set size. What's the intuition of the formula? How sensitive are your results to this choice? If the conclusions change substantially, this would be a critical limitation. If they're robust, showing this would further strengthen the paper.

**On the circularity concern:**

(3) Your definition uses the top score as a proxy for the empirical ceiling. When absolute performance is low (e.g., 40% on a hard benchmark), you call this "stagnation" or "model-level saturation." But how do we know this isn't just a genuinely hard benchmark where future breakthroughs could yield large gains? Is S_index alone sufficient to distinguish between a benchmark that is truly "model-level saturated" and one that is simply hard? If not, what additional information would help make this distinction?


**On practical impact:**

(4) The paper's recommendations are somewhat high-level. What would be the single most important change you'd recommend to a benchmark designer today, based on your findings? For example, if someone is creating a new benchmark, should they focus on making the test set larger, or on making it dynamic, or on something else entirely? A concrete, actionable takeaway would significantly increase the paper's impact on practice.

**Limitations:**

Partly, the authors have discussed limitations. They transparently address potential biases in benchmark selection, challenges with inconsistent leaderboard data, the assumption of time-invariant benchmark properties, and the restriction to text-only benchmarks. However, I would add that the paper could more explicitly acknowledge the limitation of being descriptive rather than causal (or more preferably, explore the causal mechanisms rather than the surface-level findings), and whether or not the results could be interpreted into actionable guidelines.

**Strengths And Weaknesses:**

**Strengths**

(1) **Formalization of a fuzzy concept:** The paper moves the conversation about saturation from vague, anecdotal observations to a quantifiable framework. The Saturation Index is simple, principled, and satisfies the authors' stated desiderata. This alone is a significant contribution that will enable future work.

(2) **Impressive empirical scope:** 60 benchmarks, annotated by 23 researchers with cross-checking, covering everything from AGIEval to WMT. That's a lot of work, and the resulting dataset is a contribution in itself. The forest plot in Figure 4 is clean and tells a clear story: age and test set size matter; most other things don't, once you control for confounders.

(3) **Valuable null results:** The finding that private test sets don't prevent saturation (Section 5.2) runs counter to what many people assume. The authors' explanation that "secrecy alone does not prevent compression once distributional characteristics become widely known" rings true. If you know the distribution, you can optimize for it even without seeing the exact instances. This is the kind of result that should shape how we think about benchmark design going forward. Similarly, the null effects for open-ended formats and template diversity are non-obvious and valuable.

(4) **Thoughtful framing:** The discussion of when saturation is desirable versus problematic (Section 5.4) is a highlight. Saturation reflecting genuine task mastery is progress; saturation reflecting loss of measurement resolution is a failure mode. This distinction is often elided in casual conversation about benchmarks, and having it spelled out is valuable. The separation of stagnation from saturation also clarifies thinking.

**Weaknesses**

(1) **Descriptive rather than causal:** This is the most significant limitation, and it directly addresses the user's question about impact. The paper provides strong correlational evidence, i.e. benchmark age and exposure predict saturation, but does not establish why this occurs. Is it memorization? Benchmark-specific overfitting? Distributional familiarity? Loss of novelty? The authors speculate about mechanisms but do not test them. A causal graph or interventionist approach (e.g., comparing benchmarks with similar age but different exposure patterns, or tracking how saturation changes after a paradigm shift) would strengthen the paper's ability to guide future benchmark design. As it stands, the paper tells us that saturation happens and which factors correlate with it, but not how to intervene to prevent it. This limits the paper from generating a huger impact and guiding the community on how we evaluate benchmarks.

(2) **The definition feels a bit circular:** Saturation is defined as "top models cannot be statistically distinguished AND performance approaches the empirical ceiling." But the empirical ceiling is just the top score. So a benchmark is saturated when the top models are close to the top score and indistinguishable from each other. This is basically saying "saturation is when scores are high and tightly clustered." That's fine as an operationalization, but it's not clear we've learned something deep about the benchmarks themselves—we've just observed that scores have converged. The causal claims about what drives saturation then become tricky because the outcome variable is partly defined by the scores being high. This circularity makes it difficult to separate genuine task mastery from measurement artifacts.

(3) **The effective sample size formula is underspecified:** Equation (2) defines n_eff = n^α with α=0.5 by default. This downweights large test sets to avoid "overly strong dependence of the saturation calculation on test set size." The choice of α affects the standard errors, which affect whether differences are considered statistically significant, which affects the saturation index. The paper doesn't provide any justification for this functional form or the default value. It feels like a free parameter that could substantially change results, and there's no sensitivity analysis. How sensitive are the conclusions to different values of α?


(4) **Limited actionability:** While the paper provides excellent diagnosis, the prescriptions are somewhat generic: "increase evaluation resolution," "use dynamic updates," "report uncertainty-aware statistics." These are sensible but vague. How much larger should test sets be? What kind of dynamic updates work best? The paper does not provide concrete guidance for benchmark designers beyond high-level principles. This again relates to the user's question about impact: the paper identifies problems but stops short of providing solutions that practitioners can immediately implement.

---

> ### Author Rebuttal · Authors · 2026-03-31
>
> We thank the reviewer for the detailed review and for highlighting the contributions of our work, including the formalization of benchmark saturation, the breadth of our empirical analysis, and our findings. We are glad that the reviewer found the framing, released dataset, null results, and overall scope valuable. Below, we respond to the reviewer’s questions and concerns.
>
> **Limited actionability** We thank the reviewer for pointing out this aspect; we agree that improving actionability is important. Based on our findings, there are a couple of actionable and takeaways that can guide benchmark designers.
> First, our results suggest two concrete design implications: (i)static benchmarks should be complemented with dynamic or adversarial updates to reduce exposure-driven optimization, since our analysis shows that benchmarks with these properties tend to retain discriminative power for longer and are less prone to early saturation, and (ii) benchmark providers should explicitly monitor benchmarks' saturation over time and define revision or retirement criteria once top models become statistically indistinguishable.
> Second, our findings indicate that commonly assumed safeguards, such as private test sets, open-ended formats, or template diversity, do not reliably prevent saturation and should not be relied upon as primary design strategies.
> Finally, evaluation resolution is a key actionable factor: benchmark designers should ensure test sets are large enough to resolve performance differences, for example by scaling evaluation size with model progress or verifying that score differences exceed evaluation uncertainty.
> We will incorporate these more concrete guidelines in the revised version.
>
> **Descriptive rather than casual** We agree that our analysis is observational rather than causal. Our aim is for the proposed framework and dataset to enable such studies in the future, for example by tracking saturation longitudinally or comparing benchmarks under controlled conditions. We view this work as a necessary first step toward such investigations.
> However, our results can define the direction of mechanisms to be studied: the strong correlation with benchmark age and the lack of effect for factors such as test set privacy or output format suggest that saturation is driven less by specific design choices and more by exposure and optimization pressure.
> We thank the reviewer for this point and will clarify this positioning and expand the discussion of future directions in Section 6.
>
> **Circular definition** As you have correctly pointed out, our saturation definition relies on observed score distributions on available leaderboards. However, we don't define saturation in our framework only by high performance, but by the loss of reliable separability among top models relative to evaluation uncertainty. As we discuss in Sec 5 (L344 ff.) -  based on previous findings and literature, we "imply that saturation often reflects loss of relative separability among top-performing models rather than complete task mastery." We will clarify this point more explicitly in the paper to avoid the assumption that saturation is defined by high scores.
>
> **Effective sample size** The intuition behind n_alpha is to consider test set size in the uncertainty estimate calculation while avoiding a setting where a few very large benchmarks dominate the calculation. In our dataset of 60 benchmarks, the benchmark sizes differ substantially, from 30 to 601,734 test samples, with a median of 3,000 and a highly skewed distribution due to a small number of very large benchmarks being present.
> Using n would therefore make the uncertainty term overly driven by those outliers. We chose alpha α=0.5 as midpoint to compress these size differences while still preserving the intuition that larger benchmarks should yield lower uncertainty due to the number of their samples. We agree that this choice should be made more explicit and will add both this justification and a sensitivity analysis over alpha in the revised version.
>
> **Stagnation vs saturation** We agree that distinguishing between hard benchmarks and "model-level saturation" is challenging and sometimes vague. It happens that assumed saturation  plateaus in future turn out to be stagnations rather than true saturation. History supports this: multiple benchmarks that appeared saturated at the time (such as Imagenet or GLUE-benchmarks with only minimal year-over-year improvements) later had renewed progress after architectural or paradigm shifts (e.g., large scale pretraining) were introduced.

---

> > ### Author Rebuttal · Reviewer_7yBx · 2026-04-01
> >
> > Thank you for the detailed response. I believe I'm on the same page as the authors on both the strengths and areas for future improvement of the paper. I will retain my assessment of 4 - Weak Accept for now, but will also be sure to incorporate other reviewers' evaluation and comments and adjust the score if needed.

---

> > > ### Author Response · Authors · 2026-04-03
> > >
> > > Thank you for your acknowledgment. We also wanted to use this space to address the sensitivity analysis question you and other reviewers have asked about.
> > >
> > > We agree that sensitivity regarding the selection of k and α is an important aspect. We did a sensitivity analysis varying k∈{3, 5} and α∈{0, 0.5, 1}. Across these settings, the resulting saturation indices remain highly correlated, indicating that the relative ranking of benchmarks is largely preserved. Below is an overview of correlation and the fraction of benchmarks that remain in the same bins:
> > >
> > > | Setting comparison | Spearman correlation | Same bin (%) |
> > > |-------------------------------|----------------------|--------------|
> > > | (k=3) vs (k=5) | 0.92 | 48.3% |
> > > | (\alpha=0.5) vs (\alpha=0) | 0.88 | 23.3% |
> > > | (\alpha=0.5) vs (\alpha=1) | 0.92 | 18.3% |
> > >
> > > The table above shows that we observe variation in bin assignments, which is expected given thresholding over a continuous score. However, most changes occur between neighboring bins rather than large shifts, suggesting that the underlying signal is stable even when scores vary.
> > >
> > > We further observe that a smaller k-value (e.g., k=3) increases variance due to limited model coverage, while larger k risks mixing frontier and non-frontier models given incomplete, static leaderboard data. Similarly, \alpha=1 leads to strong dependence on test set size, whereas \alpha=0 ignores evaluation uncertainty. The choice \alpha=0.5 is a balanced trade-off, moderating dataset size effects while preserving uncertainty awareness.
> > >
> > > Overall, while absolute saturation values may shift slightly, the conclusions and benchmark ordering remain stable, supporting the robustness of our framework. We will include this analysis in the revised version of the paper as well.
> > >
> > > We want to thank the reviewer for the thoroughness of their review. Your comments will help us significantly improve our paper, thank you so much!

---

### Official Review · Reviewer_65r9 · 2026-03-13

**Soundness:** 2
**Presentation:** 3
**Significance:** 2
**Originality:** 3
**Overall Recommendation:** 3
**Confidence:** 4

**Summary:**

This paper considers benchmark saturation: when benchmarks are no longer to distinguish model quality in statistically significant ways. We see this characterized often by many models grouped together at the top of a given leaderboard with very similar score, likely all within confidence bounds. The authors propose a "saturation index" to measure this more formally. They then apply this to 60 benchmarks, finding 29 of them show serious signs of saturation. The authors find the main indicator towards saturation is benchmark age. The authors also find that common methods to delay saturation, such as private held-out sets, do not seem to help in reality. The authors also suggest that expert curated benchmarks are more resistant to saturation.

**Compliance With Llm Reviewing Policy:**

Affirmed.

**Final Justification:**

While some points were addressed by the rebuttal, the most consequential comments remain unaddressed.

Mainly, the core methodology remains concerning: the author's proposed saturation index formulation is brittle w.r.t the chosen k and $\alpha$--- the effects of this brittleness propagate through the rest of the paper, weakening the main conclusions of the paper.

From the author's own reported numbers, ~80% of the benchmarks change bins when $\alpha$ is changed; $\alpha$ is a arbitrarily picked hyperparameter (as is k). That in itself would change one of the author's main claims that "Of the 60 benchmarks analyzed, 29 exhibit high or very high saturation". Some of the hypothesis test results could change too.

Finally, the confound on leaderboard density is unaddressed, copying from my rebuttal comment:  "Older benchmarks -> more models evaled -> tighter top 5 -> higher saturation index. The authors do not control for leaderboard size in their analysis."

Because the authors have been unable to address the core concerns, which in turn weaken the core findings of the paper, I do not believe the paper meets the bar for acceptance into ICML.

**Key Questions For Authors:**

How was K decided when running analysis? Did the selection of K change the paper's findings significantly? Should K be designed per benchmark? e.g I have a benchmark ranking 100 items, should I chose K of 10? what about a benchmark with 10 items; what K should I use there?

**Limitations:**

Yes.

**Strengths And Weaknesses:**

Strengths: The paper considers an important problem. A standard way to identify when a benchmark is saturation would be a great contribution.

Weakness/comments:

1. In eq 1, the authors say the SE formula is for "accuracy-like" metrics. This is never defined well. There are a decent number of scoring systems that would not work with this formula, e.g. Elo, pass@k. Can the authors give a more general method for SE estimation?

2.  There is this kind of brittle dependence on K. Can the authors resolve this or show it is not brittle in practice?

3. R_norm has weird behavior when s_1 - s_k --> 0 and presumably SE --> 0 as well. It is potentially numerically unstable in precisely in the regime where it is an important indicator.

4. The saturation index formula looks completely made up? There isn't much, if any, justification on how the authors arrived at this formulation.

5. Most of the hypotheses reduce to "how old is this benchmark". Though the paper acknowledges this, I fail to understand why we are even asking all these hypotheses when they are clearly essentially the same. Taking this away, the scope of the paper is reduces by a large margin: mostly just looking if benchmark age correlates with benchmark saturation.

6. Line 348 says that discriminative power depends on statistical resolution--- isn't this by construction based on this metric? This finding essentially restating the definition.

---

> ### Author Rebuttal · Authors · 2026-03-31
>
> We thank the reviewer for the comments, and highlighting the importance of the benchmark saturation problem and the value of standardizing how we quantify saturation.
>
> **SE formulation** While the saturation index is conceptually metric-agnostic, the operationalization we introduce in the paper focuses on accuracy-like metrics due to the availability of a simple and widely applicable uncertainty estimate. For other metrics, Eq. (1) could be replaced with metric-specific uncertainty estimates (e.g., variance from repeated evaluations). While repeated evaluation would provide a more general solution, it is not feasible in our setting, as it would require rerunning the top-5 models for each of the 60 benchmarks multiple times. We will clarify this limitation in the paper and discuss more general uncertainty estimation as future direction (Sec. 6).
> We would like to clarify that with “accuracy-like metrics, we refer to metrics that are computed as averages over a fixed set of test samples (e.g., accuracy, F1, BLEU), where uncertainty can arise from finite sample size and be approximated via standard error. We agree that the estimator in Eq. (1) does not apply to all metric types (e.g., Elo ratings, pass@k), and we will clarify this more explicitly in the paper.
>
> **Dependence on k** We introduce *k* as a general parameter and avoid hardcoding a specific cutoff - the aim was to have flexibility in defining the set of top-performing models one can use to assess saturation. Our choice reflects the following trade-off: smaller values of k can lead to unstable estimates, while larger values mix frontier models with older or less relevant ones, given incomplete leaderboard coverage/sparsity of leaderboards. Most benchmarks in our dataset report approximately 5-7 frontier models, which made k=5 a reasonable choice to ensure comparability across the 60 analysed benchmarks.. We will clarify this design decision in the paper in more detail and add a discussion on how k can be changed in settings with more complete or densely populated leaderboards.
>
> **Separate k-value per benchmark** We agree that in theory we could adapt k for each benchmark depending for example on its leaderboard size. However, in this work we fix a k-value for all analysed benchmarks to allow comparability across benchmarks. Moreover, using different k values per benchmark would make the saturation index dependent on the leaderboard structure (see more details above). We will clarify this trade-off and discuss the choice of k in the revised version in more detail.
>
> **R_norm** R_norm measures the observed performance spread relative to uncertainty, i.e., a signal-to-noise ratio. In common benchmark settings, the uncertainty term will remain non-zero due to finite test sample sizes. However, we agree that in edge cases (e.g., near-perfect or 100% accuracy), the standard error may approach zero, which could lead to numerical instability.  We thank the reviewer for pointing this out and will clarify this aspect in the paper, as well as include a small numerical stabilization in the revised version.
>
> **Intuition saturation index** Thank you for raising this concern. The main goal of this paper is to "move the conversation about saturation from vague, anecdotal observations to a quantifiable framework" as it has been also correctly pointed out by reviewer 2.
> Given that benchmark saturation is widely discussed in the literature but not quantified, our aim with this work is to propose a first framework for operationalizing and measuring saturation, i.e. as being quantified as the loss of reliable separation among top-performing models considering benchmark characteristics. The key aspects we capture is the ratio between score spread and uncertainty, which should reflect how distinguishable top models are relative to evaluation noise.
> We do not claim that this is the only way to measure saturation; rather, it is a first step toward systematically quantifying and comparing it across benchmarks. We will clarify this intuition further in the revised version.
>
> **Hypotheses** We agree with the reviewer that benchmark age is the strongest and most consistent factor in our analysis. However, our goal in testing these hypotheses was not only to identify correlates of saturation, but also to examine whether commonly assumed factors hold when considered jointly with benchmark age. Our findings indicate that widely assumed safeguards such as private test sets, open-ended formats, or template diversity do not reliably prevent saturation and therefore should not be relied upon as primary design strategies. In contrast, our results suggest that static benchmarks should be complemented with dynamic or adversarial updates to reduce exposure-driven optimization, as such benchmarks tend to retain discriminative power for longer and are less prone to early saturation. We will expand the discussion in the paper to make these insights more explicit and easier to interpret.

---

> > ### Author Rebuttal · Reviewer_65r9 · 2026-04-04
> >
> > **The rebuttal does address several concerns, however the core methodological issues remain: the author's index formulation is unjustified, the k-sensitivity is unanalyzed, and finally, the confound between leaderboard density (number of evaluated models) and benchmark age could explain the paper's main finding.**
> >
> > ---
> >
> > **SE Formulation:** Reasonably resolved via clarification in paper.
> >
> > **R_norm:** Resolved enough.
> >
> > **Hypothesis:** Valid point, the insights are useful.
> >
> > ---
> >
> > **Saturation-Index:** Still not justified. The choice of exp(−R^2_norm) isn't motivated over other monotone decreasing functions of R_norm. Additionally, pairwise separability metrics do already exist in the literature (e.g., Li et al., 2024, which is included in the authors' benchmark set) that naturally handle varying numbers of models without dependence on a fixed and arbitrary K. The authors do not discuss why their formulation is preferable to these existing approaches.
> >
> > **Weakness 6:** Unaddressed--- was not mentioned in the author's rebuttal.
> >
> > **Dependence on K:** Authors have provided an explanation on why they chose K=5. They did not show analysis on the brittleness of K, or what happens when they change K. Would their conclusions change? What if K was set proportional to the benchmark size? Should benchmark with 1000 models have the same K as one with 10? A benchmark with scores 98.1, 98.0, 97.8, 92.2, 89.5 is saturated at K=3 and fine at K=5. Moreover, the SE_\delta estimate (eq. 3) depends only on s_1 and s_k. The entire saturation index reduces to the gap between the best and Kth model. A single outlier at position K could completely alter the verdict.
> >
> > Importantly, the authors fix K=5 but don't have a canonical evaluation set, so some benchmarks, particularly older ones, have more models. These issues compound; a fixed K applied to leaderboards of varying number of models means the index is systematically biased toward finding saturation in older, more popular benchmarks.  If you draw N models from some performance distribution and consider the gap between rank 1 and rank 5, that gap will shrink as N grows--- this has nothing to do with benchmark quality/saturation, just order statistics. **This confounds with their findings**. Older benchmarks -> more models evaled -> tighter top 5 -> higher saturation index. The authors do not control for leaderboard size in their analysis.
> >
> > Moreover, the authors attempt to measure a property of the benchmark itself: "a benchmark is saturated if the evaluated models cannot be reliably distinguished." but this is really jointly dependent on the models as well. Since there is no concept of identity of independence constraints, submitting 5 models that are very similar (e.g. 5 checkpoints from the same run) instantly "saturates" the benchmark. Now, the "identical model" situation is not necessarily about literally identical models, but rather that is common for models to share training data, reward models, distillation sources, etc, producing highly correlated scores.
> >
> > For these reasons, my score remains unchanged.

---

> > > ### Author Response · Authors · 2026-04-04
> > >
> > > Thank you for your acknowledgment!
> > >
> > > We also wanted to use this space to address the **sensitivity analysis** question you and other reviewers have asked about.
> > > We agree that sensitivity regarding the selection of k and α is an important aspect. We did a sensitivity analysis varying k∈{3, 5} and α∈{0, 0.5, 1}. Across these settings, the resulting saturation indices remain highly correlated, indicating that the relative ranking of benchmarks is largely preserved. Below is an overview of correlation and the fraction of benchmarks that remain in the same bins:
> > >
> > > | Setting comparison | Spearman correlation | Same bin (%) |
> > > |-------------------------------|----------------------|--------------|
> > > | (k=3) vs (k=5) | 0.92 | 48.3% |
> > > | (\alpha=0.5) vs (\alpha=0) | 0.88 | 23.3% |
> > > | (\alpha=0.5) vs (\alpha=1) | 0.92 | 18.3% |
> > >
> > > The table above shows that we observe variation in bin assignments, which is expected given thresholding over a continuous score. However, most changes occur between neighboring bins rather than large shifts, suggesting that the underlying signal is stable even when scores vary.
> > >
> > > We further observe that a smaller k-value (e.g., k=3) increases variance due to limited model coverage, while larger k risks mixing frontier and non-frontier models given incomplete, static leaderboard data. Similarly, \alpha=1 leads to strong dependence on test set size, whereas \alpha=0 ignores evaluation uncertainty. The choice \alpha=0.5 is a balanced trade-off, moderating dataset size effects while preserving uncertainty awareness.
> > >
> > > Overall, while absolute saturation values may shift slightly, the conclusions and benchmark ordering remain stable, supporting the robustness of our framework. We will include this analysis in the revised version of the paper as well.
> > >
> > > Regarding the **saturation index formulation**, our goal was to provide a simple, smooth, and interpretable mapping from the signal-to-noise ratio R_{\text{norm}} to a bounded score. The choice of \exp(-R_{\text{norm}}^2) ensures (i) monotonicity, (ii) smooth decay, and (iii) boundedness in [0,1], which makes it convenient for comparison across benchmarks without introducing hard thresholds. We agree that other monotonic transformations are possible, and do not claim this is the only valid formulation. We will clarify this design choice and discuss alternatives (including pairwise separability metrics) in the revised version.
> > >
> > > We want to thank the reviewer for the thoroughness of their review. Your comments will help us significantly improve our paper, thank you so much!

---

### Decision · Program_Chairs · 2026-04-30

**Decision:**

Accept (regular)

**Comment:**

This paper received mixed ratings. It proposes a formal, uncertainty-aware saturation index to quantify when LLM benchmarks lose discriminative power among top-performing models, and applies it to 60 text-based benchmarks annotated along 14 properties by 23 researchers. All reviewers recognized the importance of the research question. However, Reviewer raised concerns that the saturation index formulation lacks motivation, results are sensitive to hyperparameters k and α. These were not fully resolved during rebuttal.

The AC believes these limitations do not outweigh the contribution. Rank correlations remain high across settings, indicating robust relative ordering despite bin instability. The density confound is real but the joint regression controls for adoption proxies, and test set scale emerges as an independent predictor. As Reviewer noted, this is a first systematic quantification of benchmark saturation and provides a reproducible baseline for future refinement.

Therefore, the AC recommends accepetance and encourages the authors to include the sensitivity analysis in the main paper emphasizing continuous statistics over bins, discuss and control for leaderboard density, address the circularity concern, and incorporate the representative case studies from rebuttal.